# Depth Completion as Parameter-Efficient Test-Time Adaptation

## Abstract

We introduce CAPA, a parameter-efficient test-time optimization framework that adapts pre-trained 3D foundation models (FMs) for depth completion, using sparse geometric cues. Unlike prior methods that train task-specific encoders for auxiliary inputs, which risk degrading the pre-trained prior and limit generalization to new scenes or sensor configurations, CAPA freezes the FM backbone to preserve its strong geometric prior. It updates only a small set of parameters using Parameter-Efficient Fine-Tuning (*e.g.*, LoRA or VPT), guided directly by gradients calculated from the sparse observations available at inference. This approach effectively grounds the foundation model's geometric prior in the scene-specific measurements, correcting distortions and misplaced structures. For videos, CAPA introduces sequence-level parameter sharing, jointly adapting all frames to exploit temporal correlations, improve robustness, and enforce multi-frame consistency. CAPA is model-agnostic, compatible with ViT-based depth/multi-view FMs, and achieves state-of-the-art results across diverse condition patterns on both indoor and outdoor datasets.

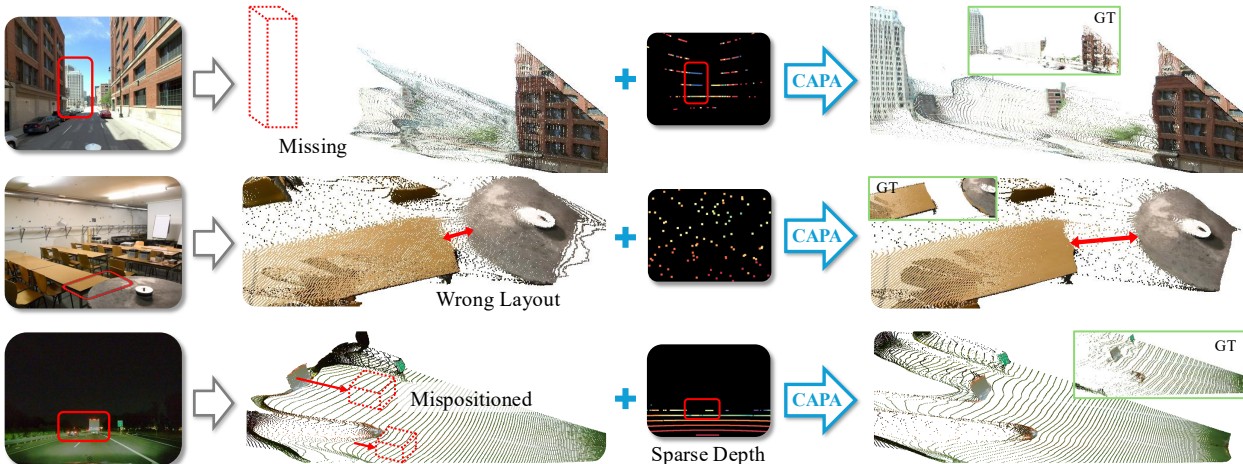

Figure 1: **CAPA performs depth completion by adapting geometric foundation models at test-time.** By aligning the strong geometric prior of a base model with the sparse depth information of test samples, one obtains accurate reconstructions of scene layout and fine details, overcoming limitations of the base model such as distorted surfaces and misplaced objects, even under challenging conditions.

## 1 Introduction

Depth completion is a fundamental capability in modern 3D computer vision. Dense, metrically accurate 3D geometry is an essential ingredient for downstream applications like mapping (Henry et al., 2014), novel view synthesis (Roessle et al., 2022; Ren et al., 2025), and the curation of high-quality 3D training datasets for autonomous systems (Caesar et al., 2020). While geometric anchors from active sensors (like LiDAR) or from Structure-from-Motion (SfM) are highly reliable, they are typically extremely sparse. Depth completion serves as the bridge to propagate these sparse anchors into complete, dense surfaces.

Traditional depth completion methods address this by explicitly feeding the sparse depth map into the model (Yan et al., 2022; Wong & Soatto, 2021) as an additional input channel. Within specific domains and controlled sensor setups, such dedicated architectures have proved highly effective and can accurately recover dense geometry. However, the scale of available RGB-D datasets to train such models is typically limited, which restricts their ability to generalize to new scenes (Chung et al., 2025). Furthermore, because it is practically impossible to curate training datasets that capture all possible geometric configurations and sensor characteristics, the task-specific encoders in these architectures tend to overfit to the specific setups of the training data.

The advent of *3D foundation models (FMs)* (Wang et al., 2025a; Yang et al., 2024b) offers a compelling solution for generalization, but these models still require proper grounding. Even the most capable FMs can exhibit defects like incorrect scaling and spatially varying distortions, making external geometric cues necessary to reliably achieve metric accuracy. However, adapting FMs via offline fine-tuning, *e.g.*, by training an extra, task-specific encoder (Wang et al., 2025e; Lin et al., 2025), risks degrading the pre-trained prior that makes them so powerful.

Test-Time Adaptation (TTA) has emerged as a promising alternative. Recent and concurrent works have explored TTA through diffusion-based models (Viola et al., 2025) or pixel-space prompting (Jeong et al., 2025). However, a critical gap remains to design a more flexible architecture that naturally generalizes to multi-frame sequences and to various sparse conditioning patterns and noise characteristics.

To address this, we propose CAPA, a flexible, high-performance framework designed for general, sensor-agnostic depth completion. Its core idea is to reframe depth completion as "Grounding FMs with geometric cues". Instead of altering the core weights, CAPA keeps the backbone frozen and updates only a compact set of new parameters using Parameter-Efficient Fine-Tuning (PEFT) techniques like LoRA (Hu et al., 2022) or VPT (Jia et al., 2022), guided by gradients calculated directly from the sparse observations available at inference time. This strategy enables effective, scene-specific adaptation during inference while preserving the global geometric understanding encoded in the original foundation model.

When extending CAPA to the video domain, the PEFT formulation naturally enables sequence-level adaptation: by sharing a common set of learnable parameters across frames of a sequence (instead of tuning each frame independently), one inherently exploits the strong correlations between neighboring frames in terms of both geometry and appearance. This aggregation of depth cues improves robustness to sparse or noisy observations and enforces temporal consistency (see Tab. 4). In practice, adaptation is performed efficiently with mini-batch updates, yielding a compact, scene-consistent calibration for all frames.

CAPA is broadly compatible and model-agnostic within the family of ViT-based depth/multi-view foundation models, applicable to any 3D foundation model with a Vision Transformer (ViT) backbone. By aligning the base model's geometric prior with the sparse depth of each test sample, CAPA produces accurate, dense reconstructions that adapt to scene-specific geometry and correct inherent distortions (see Fig. 1). Through extensive experiments on diverse datasets, CAPA consistently achieves the state-of-the-art performance with significantly lower errors than prior methods. Qualitatively (Fig. 3), this yields cleaner, more coherent depth maps and finer structural details, underscoring CAPA's strength as a practical and effective way to specialize foundation models to new scenes directly at inference time. This capability effectively targets high-fidelity offline applications like 3D mapping and pseudo-ground-truth generation. In summary, our contributions are:

- Reframing depth completion as *parameter-efficient adaptation* of foundational 3D vision models, guided by sparse depth available at test time;

- Introducing *sequence-level parameter sharing* for video depth completion, leveraging strong inter-frame correlations for improved temporal consistency and robustness;

- Conducting comprehensive evaluation and ablation studies with two PEFT strategies (LoRA and VPT) and three distinct base models (VGGT, MoGe-2, UniDepthV2);

- Achieving the best performance on four datasets captured in diverse environments, with varying condition patterns.

## 2 Related Work

### 2.1 Depth estimation

Monocular depth estimation is a dense regression problem that has advanced rapidly over the past decade. Pioneer work by Eigen et al. (Eigen et al., 2014) laid the foundation, followed by methods exploring ordinal regression (Fu et al., 2018), depth binning (Bhat et al., 2021; 2023), canonical camera-space representations (Yin et al., 2023), *etc.* A major step forward was improving generalization through training on large-scale datasets (Ranftl et al., 2020; 2021). More recently, foundation models trained on massive and diverse data (Yang et al., 2024a;b; Wang et al., 2025b;c) or repurposed from diffusion-based image generators (Ke et al., 2024; 2025b; Fu et al., 2024) have achieved state-of-the-art zero-shot performance across a wide range of scenes.

The foundation model paradigm has also extended to video depth estimation. Earlier approaches primarily relied on optical flow and pose optimization to enforce temporal consistency (Luo et al., 2020; Zhang et al., 2021). More recent methods, built upon strong pre-trained priors (Rombach et al., 2022; Oquab et al., 2023), such as RollingDepth (Ke et al., 2025a) and VideoDA (Chen et al., 2025), achieve robust generalization and temporal coherence without the need for explicit post-processing.

In parallel, feed-forward 3D reconstruction models have emerged, jointly predicting multiple geometric outputs. This line of work was pioneered by DUSt3R (Wang et al., 2024), which predicts point maps from unposed image pairs, requiring post-processing to extract camera parameters. Methods like VGGT (Wang et al., 2025a), Fast3R (Yang et al., 2025), and $\pi^3$ (Wang et al., 2025d) extended this to the multi-view setting, leveraging large-scale multi-task training to directly predict depth, camera poses, and intrinsics. Subsequent work, such as MapAnything (Keetha et al., 2025), further enhanced this direction by conditioning on available geometric cues. Our proposed CAPA builds upon these strong single-frame or multi-view geometric foundation models, enhancing them through our test-time adaptation framework.

### 2.2 Parameter-Efficient Fine-Tuning

The vast scale of Foundation Models has driven the development of Parameter-Efficient Fine-Tuning (PEFT) methods, which adapt large models by freezing the backbone and updating only a small, additive or reparameterized set of parameters (Han et al., 2024). We focus on two primary types when adapting attention (Vaswani et al., 2017) layers: Low-Rank Adaptation (LoRA) (Hu et al., 2022), which updates projection matrices via low-rank decomposition, and Visual Prompt Tuning (VPT) (Jia et al., 2022), which injects learnable tokens into the input token sequence. The technical details are provided in Sec. 3.2.

In addition to LLM tasks, PEFT has recently been explored in the 3D domain for scene-specific calibration. For instance, LoRA3D (Lu et al., 2025) self-calibrates foundation models via low-rank adaptation using pseudo-labels derived from multi-view alignment, and Test3R (Yuan et al., 2025) utilizes visual prompt tuning optimized through a self-supervised geometric consistency loss across frame triplets. In this work, we investigate PEFT for depth completion, assuming the availability of sparse geometric cues, and frame the task as a test-time adaptation of pre-trained depth models.

### 2.3 Depth Completion

Early methods approached depth completion as an image-guided dense regression task, with Ma & Karaman (2018) introducing a sparse-to-dense CNN framework to predict dense depth from sparse samples. Subsequent supervised methods significantly improved performance by propagating sparse measurements via spatial propagation networks (SPNs) (Cheng et al., 2019; 2020; Park et al., 2020), affinity kernels (Imran et al., 2019), or bilateral propagation (Tang et al., 2024). Later approaches expanded receptive fields and captured non-local structures using graph neural networks (Xiong et al., 2020), hybrid CNN-Transformer architectures (Zhang et al., 2023), or recurrent optimization-guided frameworks like OGNI-DC (Zuo & Deng, 2024) and its multi-scale variant OMNI-DC (Zuo et al., 2025). More recent dedicated networks have further enhanced structural accuracy through semantic assistance (Yan et al., 2025) or 3D view decomposition (Yan et al., 2024). Distinct from purely supervised learning on limited datasets, other works explored unsupervised

paradigms by explicitly grounding completion in multi-view geometry or photometric consistency to bypass the need for dense ground truth. For instance, Wong *et al.* explicitly modeled 3D topology via scaffolding derived from Visual Inertial Odometry (Wong et al., 2020) and enforced intrinsic geometric constraints through calibrated backprojection layers (Wong & Soatto, 2021).

Recent offline training-based strategies leverage pre-trained priors by fusing features with sparse depth (*e.g.*, PriorDA (Wang et al., 2025e) and PromptDA (Lin et al., 2025)) or adapting diffusion models (Gui et al., 2025). Attempts to achieve sensor-agnostic capabilities include MapAnything (Keetha et al., 2025), which trains a unified multi-task model, and UniDC (Park & Jeon, 2024), which explores hyperbolic embeddings. Despite improved performance, offline training or finetuning still risks degrading the generic pre-trained prior that underpins broad generalization.

Finally, test-time adaptation methods refine predictions directly at inference. Diffusion-based approaches such as Marigold-DC (Viola et al., 2025) and related works (Hyoseok et al., 2025; Gregorek & Nalpantidis, 2025) update latent features using gradients from the sparse supervision. Other recent approaches adapt pre-trained depth completion models using sparse depth proxies (ProxyTTA (Park et al., 2024)), learned energy functions (ETA (Chung et al., 2025)), or domain-specific embeddings for unsupervised continual adaptation (ProtoDepth (Rim et al., 2025)). Unlike ProxyTTA and ETA, which update the full weights of specialized, domain-specific depth completion networks or rely on auxiliary energy models, CAPA adapts 3D foundation models without altering their core weights via PEFT. A concurrent effort, TestPromptDC (Jeong et al., 2025), pursues a similar TTA goal but operates in image space with pixel-level prompts. This design is sensitive to noise and not well-suited for multi-frame inputs. In contrast, our framework enables robust scene-specific adaptation precisely by sharing visual prompts or low-rank adapters internally across multi-frame sequences, a key capability for video consistency missing in prior TTA works.

## 3 Method

**Problem statement.** Given an input RGB image $\mathbf{I} \in \mathbb{R}^{H \times W \times 3}$ and a corresponding sparse depth map $\mathbf{C} \in \mathbb{R}^{H \times W}$ (where valid measurements are indicated by a mask $\mathbf{M} \in \mathbb{R}^{H \times W}$), *depth completion* is to predict the complete, dense depth map $\hat{\mathbf{D}} \in \mathbb{R}^{H \times W}$ that is consistent with the sparse input cues. We further extend this to the multi-view or video setting, where $N$ frames and depth maps are jointly used to produce geometrically and temporally consistent predictions $(\hat{\mathbf{D}}_1, \ldots, \hat{\mathbf{D}}_N)$.

### 3.1 Foundation Model Backbone

Many recent advanced 3D foundation models, like DepthAnything (Yang et al., 2024a), MoGe (Wang et al., 2025b), and VGGT (Wang et al., 2025a), are built on Vision Transformer (ViT) image encoders, most commonly DINOv2 (Oquab et al., 2023), which provides strong geometric and semantic priors, obtained through self-supervised pretraining. Typically, these encoders divide an input image into a sequence of patch tokens representing local image features, which are then processed through multiple self-attention layers. Rather than designing task-specific networks, CAPA targets these pre-trained encoders directly. We primarily use VGGT (Wang et al., 2025a) to instantiate our adaptation approach. In its pipeline, tokens from DINOv2 are processed by a multi-view aggregator and decoded into dense depth maps via a DPT (Ranftl et al., 2021) head. By updating only the attention layers within the ViT, we can guide the model's global geometry to align with sparse observations, avoiding the need to fine-tune the dense decoder or the full backbone. Intuitively, adaptation of the encoder seems to suit the task better than adapting only the head, because sparse depth cues often point to global geometric biases, *e.g.*, incorrect scale, distorted layout, wrong placement of objects or a lack of temporal consistency. Adapting attention layers can modulate token interactions and help propagate sparse cues through the frozen geometric prior. This intuition is consistent with Tab. 7, where tuning the full encoder outperforms tuning only the head, and the performance of encoder-side LoRA/VPT is close to full fine-tuning. While we describe our framework primarily using VGGT, this parameter-efficient adaptation is model-agnostic and can easily transfer to other ViT-based foundation models, including single-view architectures (see Fig. 5).

### 3.2 Parameter-Efficient Fine-Tuning

In this section, we introduce two complementary PEFT techniques designed to change the final attention outputs with a minimal number of learnable parameters. We first recap the core attention mechanism (Vaswani et al., 2017), focusing on the scaled dot-product attention formulation for clarity.

Given an input token sequence $\mathbf{X} \in \mathbb{R}^{L \times d_c}$, where $L$ is the token length and $d_c$ the channel dimension, the attention update is:

$$\text{Attention}(\mathbf{X}) = \text{Softmax}\left(\frac{\mathbf{Q}'\mathbf{K}'^T}{\sqrt{d_k}}\right)\mathbf{V}', \tag{1}$$

where queries, keys, and values are obtained via learned projections $\mathbf{W}_q, \mathbf{W}_k, \mathbf{W}_v \in \mathbb{R}^{d_c \times d_k}$ ($d_k$ being the projected channel dimension):

$$\mathbf{Q}' = \mathbf{X}\mathbf{W}_q, \quad \mathbf{K}' = \mathbf{X}\mathbf{W}_k, \quad \mathbf{V}' = \mathbf{X}\mathbf{W}_v.$$

**Low-rank adaptation.** LoRA (Hu et al., 2022) adapts the model by augmenting the projection matrices $\mathbf{W}_m$ ($m \in \{q, k, v\}$) with low-rank updates. The key assumption is that the task-specific weight update $\Delta\mathbf{W}_m$ lies in a low-dimensional subspace:

$$\mathbf{W}'_m = \mathbf{W}_m + \Delta\mathbf{W}_m, \quad \Delta\mathbf{W}_m = \mathbf{B}_m\mathbf{A}_m, \tag{2}$$

where $\mathbf{B}_m \in \mathbb{R}^{d_c \times r}$ and $\mathbf{A}_m \in \mathbb{R}^{r \times d_k}$ and $r \ll \min(d_c, d_k)$. During adaptation, only $\mathbf{A}_m$ and $\mathbf{B}_m$ are trained, while $\mathbf{W}_m$ remain frozen. This enables LoRA to alter attention projections through a compact low-rank update, efficiently adjusting the attention response.

**Visual prompt tuning.** VPT (Jia et al., 2022) instead modulates the attention computation by expanding the token sequence. At each transformer layer, a small set of trainable *prompt tokens* $\mathbf{P} \in \mathbb{R}^{t \times d_c}$ is prepended to the input sequence:

$$\mathbf{X}_{\text{new}} = [\mathbf{P}; \mathbf{X}]. \tag{3}$$

The attention is then computed over the extended token sequence, where the trainable prompt tokens modulate the attention map and thus influence how image tokens attend to one another. After the attention layer, only the updated image tokens are retained and passed forward, preserving the original token lengths.

**In summary**, both LoRA and VPT achieve parameter-efficient model adaptation by manipulating the same component—attention. LoRA modifies the *projection layers* to alter the projected token values directly, whereas VPT injects *learnable tokens* to reshape the attention distribution implicitly. While they offer two complementary mechanisms, their effect is similar: modulate the attention to customize a pre-trained model with task-specific cues.

### 3.3 Depth Completion as Model Adaptation

**Model adaptation.** Given an input pair $(\mathbf{I}, \mathbf{C})$, we adapt the pre-trained model $\mathbb{F}$ to sparse depth measurements by iteratively optimizing a minimal set of parameters $\theta$, corresponding to either LoRA's low-rank projection matrices $(\mathbf{A}_m, \mathbf{B}_m)$ or VPT's prompt tokens $\mathbf{P}$.

As illustrated in Fig. 2, at each iteration, the adapted model $\mathbb{F}_\theta$ predicts a dense depth map $\hat{\mathbf{d}}$. To resolve the scale ambiguity inherent in foundation models, we first compute an affine transformation (scale $s$ and shift $t$) that aligns the raw prediction $\hat{\mathbf{d}}$ with the sparse measurements $\mathbf{C}$ by solving the robust L1 minimization:

$$\min_{s,t} \left| \mathbf{M} \odot \left( s \cdot \hat{\mathbf{d}} + t - \mathbf{C} \right) \right|_1 \tag{4}$$

The resulting aligned depth, $\hat{\mathbf{D}} = s \cdot \hat{\mathbf{d}} + t$, is then used to compute the L1 loss against $\mathbf{C}$ at valid pixels, which is subsequently backpropagated to update the parameters $\theta$.

**Sequence-level adaptation.** For video or multi-frame inputs, we propose a sequence-level adaptation strategy by sharing the same set of trainable parameters $\theta$ across all frames in a sequence. This design is founded on the observation that frames within the same scene exhibit consistent global geometric and texture characteristics. Consequently, the pre-trained foundation model exhibits highly correlated behavior when processing these frames, allowing a single, shared set of $\theta$ to effectively capture the necessary scene-specific adjustments for the entire sequence. This approach aggregates sparse geometric cues from multiple views, leading to two main benefits: it significantly improves robustness against imperfect or noisy condition points, and enhances geometric consistency across multi-view or temporal frames. Compared to per-frame optimization, optimizing over diverse viewpoints within the scene stabilizes the optimization process by averaging out noise and local errors present in individual frame gradients, ensuring that the learned $\theta$ represents a consistent scene-specific calibration. To efficiently handle long sequences, we employ a mini-batch strategy: at each optimization step, a random subset of frames is sampled to compute the loss and update $\theta$. This strategy substantially reduces the computational overhead compared to processing the entire sequence simultaneously at each step.

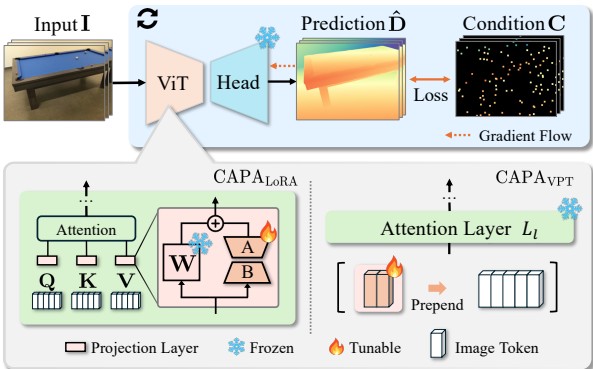

Figure 2: **Method overview of CAPA.** CAPA adapts 3D foundation models given sparse conditional depth (**C**) by efficiently tuning its image encoder while keeping all pre-trained weights frozen. This is achieved by manipulating the attention layers via two methods: 1) CAPA$_{\text{LoRA}}$, which adds low-rank adapters to the projection weights, or 2) CAPA$_{\text{VPT}}$, which prepends tunable prompt tokens to the image token sequence before each attention layer.

## 4 Experiments

### 4.1 Experimental Settings

**Evaluation datasets.** We primarily evaluate our method on four datasets, covering both indoor and outdoor environments, specifically selected to provide high-quality, consistent, dense ground truth depth and diverse testing conditions. ScanNet (Dai et al., 2017) is an indoor RGB-D video dataset that provides globally consistent depth annotation. We use the 100 sequences in the test split for evaluation. 7-Scenes (Glocker et al., 2013) is an indoor RGB-D video dataset (18 sequences), with challenging camera motions to test temporal robustness. For these two datasets, we uniformly sample 100 frames within the first 300 ones of each video. iBims (Koch et al., 2018) is a single-image indoor dataset containing 100 challenging RGB-D samples with scanner-level accurate depth annotations, widely used in depth estimation and completion tasks. Mapillary Metropolis (Antequera et al., 2020) is a city-scale outdoor dataset, providing survey-grade precise and dense annotations, avoiding specific sparsity and bias. We use its validation split, which includes 36 sequences with 68 to 96 frames (after discarding one sequence for which the ground-truth depth is partially missing).

**Sparse depth patterns.** We experiment with different point patterns for 3D conditioning, reflecting diverse possible application scenarios : (i) *keypoints,* where we sample SIFT (Lowe, 2004) keypoints on ScanNet and iBims and use theSfM points from COLMAP (Schonberger & Frahm, 2016; Brachmann et al.,

Table 1: **Quantitative comparison of CAPA with baseline methods** [AbsRel (%)↓]. The second header row reports the number of frames/images per evaluated sample.

| | ScanNet 100 frames | | | 7-Scenes 100 frames | | | iBims single image | | | Metropolis 68–96 frames | | | Avg. Rank |
|---|---|---|---|---|---|---|---|---|---|---|---|---|---|
| | SIFT | 100 | $< 3m$ | SfM | 100 | $< 3m$ | SIFT | 100 | $< 5m$ | 8-line | 16-line | 32-line | |
| DepthAnythingV2 | 4.8 | 3.8 | 4.7 | 5.4 | 4.6 | 4.6 | 4.0 | 3.3 | 9.2 | 54.0 | 57.0 | 56.2 | 9.8 |
| UniDepthV2 | 3.2 | 2.7 | 2.8 | 4.2 | 3.7 | 3.7 | 3.4 | 2.7 | 3.2 | 29.4 | 28.3 | 27.7 | 6.6 |
| MoGe-2 | 4.1 | 3.3 | 3.4 | 3.9 | 3.3 | 3.3 | 3.4 | 2.7 | 3.1 | 22.1 | 20.7 | 20.2 | 6.1 |
| VGGT | 2.2 | 2.0 | 2.0 | 4.7 | 4.2 | 4.2 | 4.0 | 3.3 | 3.9 | 10.2 | 10.2 | 9.9 | 5.5 |
| VideoDA | 5.0 | 4.1 | 4.2 | 6.1 | 5.1 | 5.1 | 5.2 | 4.0 | 4.6 | 50.8 | 47.3 | 46.5 | 10.0 |
| PromptDA | 16.6 | 13.2 | 12.2 | 8.2 | 6.9 | 4.0 | 11.6 | 7.8 | 8.0 | 27.9 | 19.6 | 16.5 | 10.1 |
| PriorDA-v1.1 | 3.4 | 2.5 | 10.3 | 3.7 | 3.1 | 9.1 | 2.9 | 2.6 | 14.9 | 26.3 | 19.5 | 15.6 | 6.8 |
| OMNI-DC-v1.1 | 2.0 | 1.5 | 5.4 | 3.3 | 2.2 | 4.6 | 1.7 | **1.5** | 10.8 | 14.5 | 11.4 | 9.1 | 4.8 |
| Marigold-DC | 4.3 | 3.7 | 2.2 | 4.1 | 4.0 | 1.6 | 7.6 | 6.1 | 5.2 | 34.0 | 25.4 | 20.0 | 7.8 |
| TestPromptDC | 4.4 | 3.8 | 2.8 | 3.1 | 4.0 | 2.6 | 7.0 | 5.1 | 5.1 | 21.8 | 18.9 | 16.9 | 6.8 |
| CAPA$_{\text{LoRA}}$ | **1.0** | **0.9** | 1.1 | 2.8 | **1.0** | **0.9** | 1.7 | 2.1 | **2.0** | **7.8** | **7.1** | **6.7** | **1.4** |
| CAPA$_{\text{VPT}}$ | 1.1 | 1.0 | **1.0** | **2.6** | 1.1 | 1.1 | **1.4** | 1.7 | **2.0** | 8.2 | 7.6 | 7.4 | 1.7 |

2021) for 7-Scenes ; (ii) *random points*, where we uniformly sample 100 valid ground-truth depth pixels over the image; (iii) *limited-range points*, where we follow PriorDA (Wang et al., 2025e) to mimic depth sensors with *limited range* and retain only points within 3 m for ScanNet/7-Scenes, respectively 5 m for iBims; and (iv) *LiDAR-like scan lines*, where we simulate 8-, 16-, and 32-line automotive LiDAR scanning patterns on Metropolis. Additionally, following OMNI-DC (Zuo et al., 2025) , we corrupt 10% of the condition points with noise , except for 7-Scenes SfM, whose reconstructed points are already affected by actual reconstruction noise. Further details and noise-free results are provided in Appendix Secs. B.2, B.3 and C.2.

**Evaluation metrics.** Following Wang et al. (2025e), we report AbsRel as the main depth metric (MAE and RMSE are in the appendix). We measure temporal consistency using the optical flow–based warping loss (OPW) (Wang et al., 2022; Ke et al., 2025a). Detailed definitions are provided in Sec. B.4.

**Baselines.** CAPA$_{\text{LoRA}}$ and CAPA$_{\text{VPT}}$ denote the two variants of our proposed method on top of VGGT (Wang et al., 2025a) base model. We compare against two groups of methods, differentiated by whether they use sparse depth input. The first group are methods that do *not* use sparse depth. Among them, DepthAnythingV2 (Yang et al., 2024b), UniDepthV2 (Piccinelli et al., 2025), and MoGe-2 (Wang et al., 2025c) are *single-frame* depth estimators; on sequence datasets, these methods are run independently frame-by-frame. VGGT (Wang et al., 2025a) is a *multi-view* reconstruction model, and VideoDA (Chen et al., 2025) is a *video-based* depth estimation model. For all these methods, we align predictions to the sparse depth observations in postprocessing, on a frame-by-frame basis. Second, we compare against depth completion methods that do take sparse depth as additional input besides images. PromptDA (Lin et al., 2025), OMNI-DC (Zuo et al., 2025), and PriorDA (Wang et al., 2025e) are feed-forward single-frame depth completion methods, while Marigold-DC (Viola et al., 2025) and TestPromptDC (Jeong et al., 2025) are test-time adaptation, single-frame depth completion methods. All these depth completion baselines are run frame-by-frame on sequence datasets. For the single-image iBims dataset, all methods are evaluated per image.

**Implementation details.** We inject learnable parameters into all 24 attention layers of VGGT's ViT encoder. We set the LoRA (Hu et al., 2022) rank to $r = 4$ and the VPT (Jia et al., 2022) prompt length to $t = 16$. Both variants yield a highly compact set of 0.39M trainable parameters. We tune each sample for 100 steps using the AdamW optimizer with a learning rate of $10^{-3}$ for LoRA and $2 \times 10^{-4}$ for VPT. The PEFT parameters are reset between test samples. We provide more details in Appendix Sec. A.

### 4.2 Main Results

**Comparison to state of the art.** Quantitative results are summarized in Tab. 1. The error metrics clearly highlight the effectiveness of our adaptation strategy. The two variants, CAPA$_{\text{LoRA}}$ and CAPA$_{\text{VPT}}$, achieve comparable performance and consistently outperform all baselines, securing a rank of first or second in every setting across four datasets. Notably, the prediction errors of CAPA are less than half as large as those of competing depth completion schemes in most cases. Furthermore, adaptation significantly boosts the base model: the VGGT error reduces by 2–3× when enhanced with CAPA.

Among depth completion methods, OMNI-DC comes closest to CAPA, but suffers from a substantial accuracy drop in the limited-range setting, likely because it is not trained for that scenario. A similar degradation is observed for other methods, such as PromptDA and PriorDA. In contrast, test-time adaptation approaches like Marigold-DC and TestPromptDC, which also adapt pre-trained models at test time, exhibit greater robustness, supporting the effectiveness of foundation models as priors. However, these methods can still be sensitive to noisy conditions. We illustrate this vulnerability in Appendix Fig. S4.

Qualitative comparisons are displayed in Fig. 3 and further extended in Appendix Fig. S5.

**Improved temporal consistency.** Temporal consistency evaluation is shown in Tab. 2 and the qualitative results are presented in Fig. 4. CAPA achieves the lowest OPW error in both indoor and outdoor settings, demonstrating the benefit of test-time adaptation.

Table 2: **Temporal consistency evaluation** [OPW(%) ↓].

|  | ScanNet | | | 7-Scenes | | | Metropolis | | |
|---|---|---|---|---|---|---|---|---|---|
|  | SIFT | 100 | $< 3m$ | SfM | 100 | $< 3m$ | 8-line | 16-line | 32-line |
| DepthAnythingV2 | 4.3 | 3.6 | 7.4 | 3.5 | 3.5 | 3.0 | 1296.6 | 1419.9 | 1413.8 |
| UniDepthV2 | 2.7 | 2.5 | 2.5 | _2.9_ | _2.9_ | 2.6 | 165.2 | _152.2_ | 141.1 |
| MoGe-2 | 4.1 | 3.7 | 3.6 | 3.1 | _2.9_ | 2.7 | 219.3 | 205.3 | 199.1 |
| VGGT | _2.3_ | _2.2_ | _2.0_ | 3.1 | 3.2 | 2.9 | 149.4 | 152.5 | 149.0 |
| VideoDA | 3.3 | 3.2 | 2.8 | 3.3 | 3.6 | 2.9 | 179.3 | 160.4 | 141.5 |
| PromptDA | 11.2 | 12.1 | 3.8 | 6.3 | 10.3 | 3.5 | 206.4 | 199.2 | 179.8 |
| PriorDA-v1.1 | 5.0 | 4.7 | 3.7 | 3.5 | 4.9 | 4.1 | 268.9 | 233.2 | 195.0 |
| OMNI-DC-v1.1 | 3.2 | 3.0 | 9.2 | 3.3 | 3.7 | 10.0 | 171.2 | 174.5 | 171.3 |
| Marigold-DC | 8.1 | 9.0 | 2.6 | 4.8 | 8.9 | 3.0 | 257.7 | 213.3 | 174.7 |
| TestPromptDC | 8.9 | 9.3 | 2.3 | 3.7 | 9.1 | _2.5_ | 185.6 | 153.9 | 138.4 |
| CAPA$_{\text{LoRA}}$ | **1.8** | **1.8** | **1.9** | **2.4** | **2.3** | **2.3** | **119.4** | **120.0** | **118.4** |
| CAPA$_{\text{VPT}}$ | **1.8** | **1.8** | **1.9** | **2.4** | **2.3** | **2.3** | _119.8_ | **120.0** | _118.6_ |

**Adherence to conditioning vs. generalization.** To analyze how different methods respond to the provided 3D conditioning, we separately evaluate the depth predictions at the conditioning points and at all other locations. As shown in Tab. 3, the errors of most depth completion schemes are significantly lower at the sparse, observed conditioning points, and increase as one moves into the parts that are actually completed. For example, for OMNI-DC, the increase is nearly 4× for ScanNet and 2× for 7-Scenes, indicating overfitting to the provided condition points. With CAPA the performance gap is much smaller—for ScanNet barely noticeable—as test-time adaptation effectively grounds the strong geometric prior of the foundation model and precisely calibrates it using the sparse condition points (see first sample in Fig. 3).

**Other base models.** We apply CAPA to two more base models, UniDepthV2 (Piccinelli et al., 2025) and MoGe-2 (Wang et al., 2025c), to demonstrate its versatility. Results, averaged over ScanNet and 7-Scenes, are shown in Fig. 5. For UniDepthV2, we also compare with TestPromptDC, which is also based on prompt tuning and uses the same base model, but injects learnable tokens into each pixel before encoding. This design leads to worse performance, likely due to noise being introduced around the condition points. In contrast, our method consistently improves the base model in terms of both depth accuracy and temporal

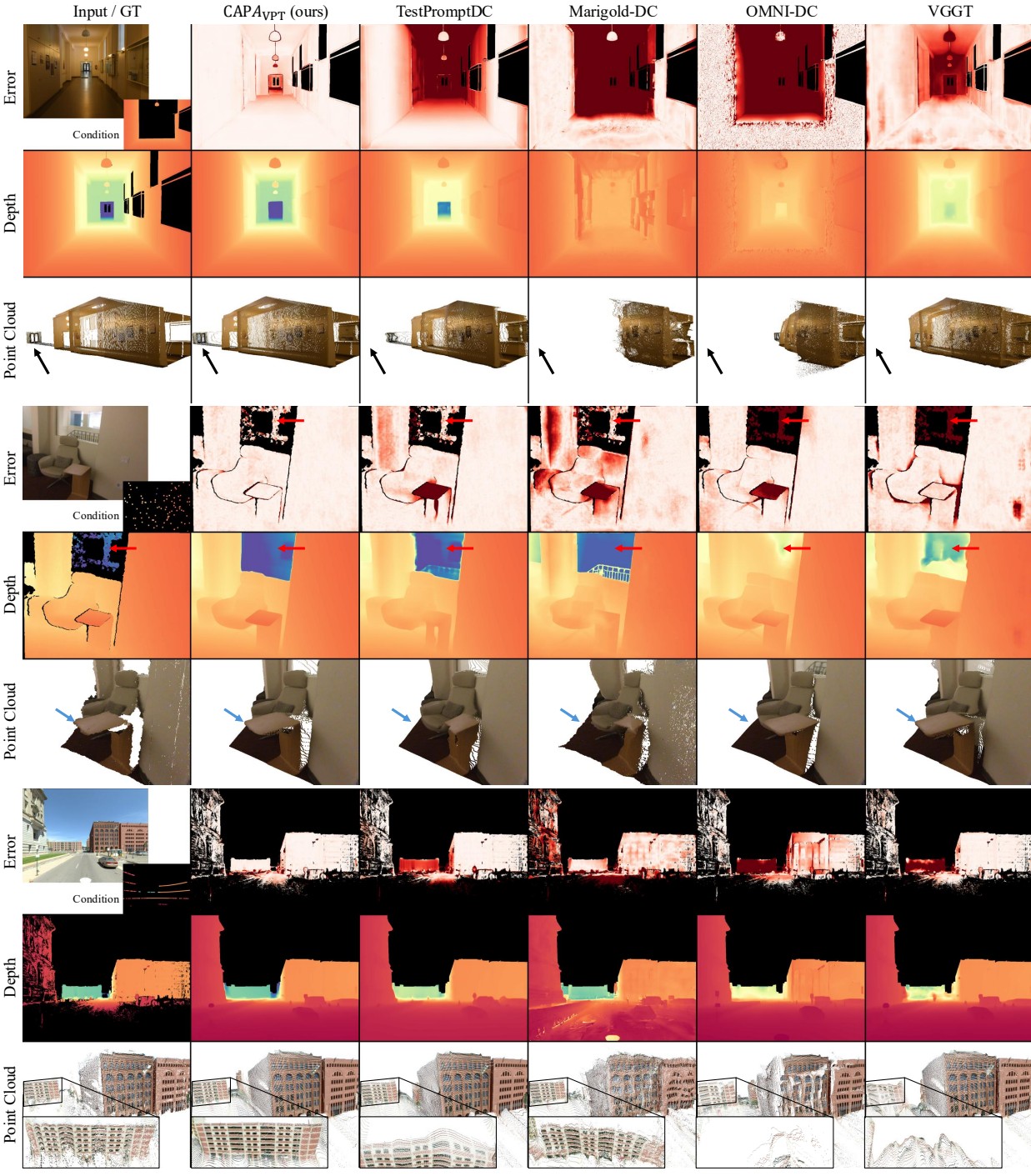

Figure 3: **Qualitative comparison on iBims, ScanNet, and Metropolis datasets**. CAPA reliably recovers the full scene: in the first sample, despite 3D points being available only within $< 5\,\mathrm{m}$, the geometry prior is calibrated sufficiently to reconstruct the full depth range, whereas most baselines focus on the well-constrained near-field; in the second sample, with only two observed points in the far-field, CAPA corrects the global geometry while preserving local structures; in the third sample, geometric structure is correctly recovered both near and far from the camera. Depth is color-coded near ▬▬ far, errors low ▬▬ high. Colored arrows mark corresponding locations across images.

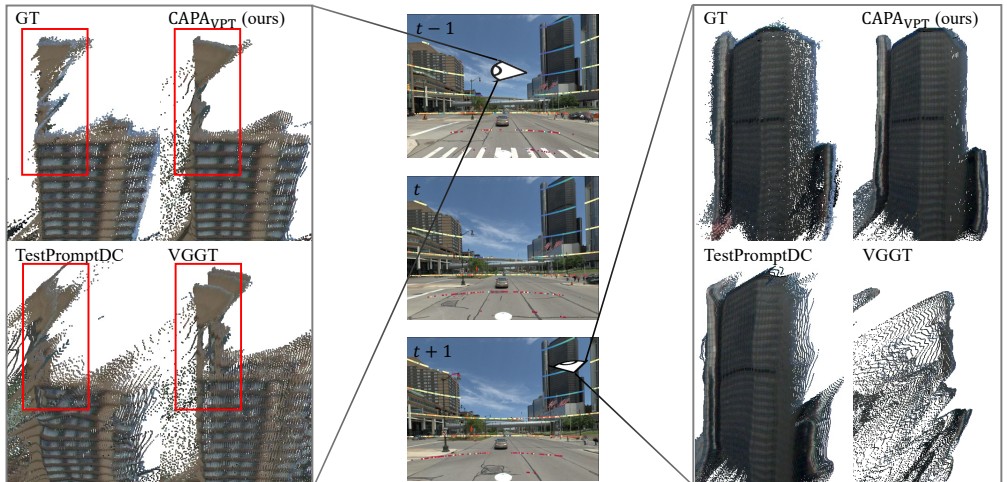

Figure 4: **Qualitative comparison of temporal consistency.** Three consecutive frames are *overlaid* using ground truth poses. On the left, CAPA reconstructs coherent building shapes, while the baselines are visibly distorted.

Table 3: **Comparison of depth errors inside and outside conditioned regions** [AbsRel (%)↓].

|  | ScanNet (100) | | | 7-Scenes (SfM) | | |
|---|---|---|---|---|---|---|
|  | In Cond. | Out Cond. | Diff. | In Cond. | Out Cond. | Diff. |
| PromptDA | 11.8 | 13.2 | 1.4 | 3.0 | 8.3 | 5.3 |
| PriorDA-v1.1 | 1.9 | 2.5 | 0.6 | 2.0 | 3.7 | 1.7 |
| OMNI-DC-v1.1 | **0.4** | 1.5 | 1.1 | 1.7 | 3.3 | 1.6 |
| Marigold-DC | 2.4 | 3.7 | 1.3 | 1.8 | 4.1 | 2.3 |
| TestPromptDC | 3.1 | 3.8 | 0.7 | 1.7 | 3.1 | 1.4 |
| CAPA$_{\text{LoRA}}$ | 0.8 | **0.9** | **0.1** | **1.6** | 2.8 | 1.2 |
| CAPA$_{\text{VPT}}$ | 0.9 | 1.0 | **0.1** | 1.7 | **2.6** | **0.9** |

consistency. Across all three base models, both LoRA and VPT reduce the depth error by $\approx 2\times$, emphasizing the general validity of our adaptation scheme beyond a specific choice of base model.

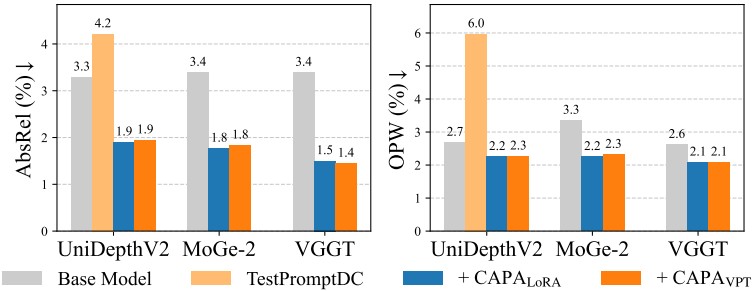

Figure 5: **CAPA results when applied to other base models**.

### 4.3 Ablation Experiments

**Datasets.** We randomly sample 32, 8, and 16 sequences from ScanNet, 7-Scenes, and Metropolis, respectively, for the ablation experiments.

**Parameter sharing within sequence.** For video depth completion, our method jointly optimizes over an entire sequence, sharing a single set of learnable parameters across all frames. As shown in Tab. 4, sequence-level adaptation reaches better depth accuracy and temporal consistency than per-frame tuning, particularly when the conditioning is sparse. We hypothesize that the gain arises because shared parameters encourage the adaptation to aggregate information across frames, and to suppress noise in the 3D condition. Besides the gain in performance, parameter sharing is also computationally more efficient: model adaptation needs to be done only once per sequence for a fixed number of steps (using mini-batch optimization), then the tuned model can be applied to all frames. We note that, as more condition points are added, per-frame tuning benefits more from the increased data density, due to the larger effective model capacity. See Appendix Sec. C.4 for further analysis with varying densities of the conditioning points.

Table 4: **Frame-level vs. sequence-level adaptation**.

|  | Parameter Sharing | ScanNet (100) | | 7-Scenes (SfM) | | Metropolis (8-line) | |
|---|---|---|---|---|---|---|---|
|  |  | AbsRel↓ | OPW↓ | AbsRel↓ | OPW↓ | AbsRel↓ | OPW↓ |
| CAPA$_{\text{LoRA}}$ | Frame | 1.7 | 4.9 | 4.3 | 4.3 | 8.7 | 153.0 |
|  | Sequence | 0.9 | 1.9 | 3.3 | 2.7 | 7.7 | 117.0 |
| CAPA$_{\text{VPT}}$ | Frame | 1.5 | 4.1 | 3.6 | 4.0 | 8.8 | 149.9 |
|  | Sequence | 1.0 | 1.9 | 2.7 | 2.5 | 8.2 | 119.0 |

**Mini-batch tuning.** For video inputs, we randomly sample a subset of frames at each optimization step to form a mini-batch for adapting the shared model. Increasing the number of frames per batch generally boosts performance, as the model is exposed to a broader range of viewpoints and thus receives more informative and stable gradients; at the cost of additional computation. As shown in Fig. 6, using just 2% of the available frames already delivers a clear improvement over the single-frame (1%) baseline, while the performance gain gradually saturates beyond $\approx 10\%$.

**Accuracy/efficiency trade-off.** The number of optimization steps determines the trade-off between quality and computational cost. As Fig. 7 shows, a few update steps already yield a noticeable improvement, with performance stabilizing after $\approx 100$ steps. This allows a flexible efficiency/accuracy trade-off : users can stop earlier when latency matters, or they can run the full schedule when reconstruction quality is the priority, a flexibility not offered by fixed-cost diffusion models.

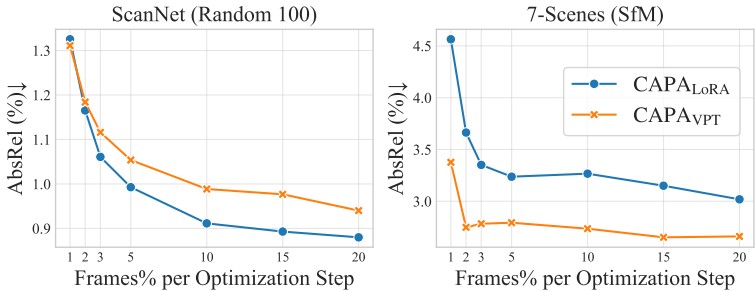

Figure 6: **Influence of mini-batch size**. Performance improves with more frames and saturates at $\approx 10\%$.

In Tab. 5, we report the total wall-clock time required to process a 100-frame video sequence from ScanNet. While CAPA is slower than a single forward pass through VGGT because of test-time optimization, it remains more than $10\times$ faster than optimization-based depth-completion baselines such as Marigold-DC or TestPromptDC. Furthermore, analyzing the interaction with mini-batch size (Tab. 6), we conclude that for a fixed budget, more optimization steps (frequent parameter updates) are generally more effective than larger frame mini-batches.

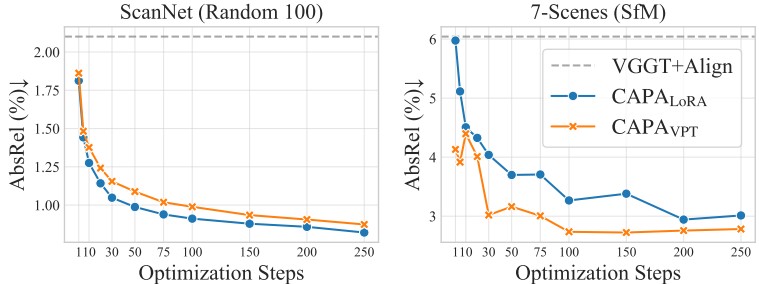

Figure 7: **Reconstruction quality vs. optimization steps.**

Table 5: **Runtime** on a 100-frame ScanNet video [sec.]

| Method | Optimization time | Inference Time | Total Time |
|---|---|---|---|
| VGGT | - | 9 | 9 |
| Marigold-DC | 1615 | - | 1615 |
| TestPromptDC | 2631 | 40 | 2671 |
| CAPA$_{\text{LoRA}}$ | 140 | 9 | 149 |
| CAPA$_{\text{VPT}}$ | 146 | 9 | 155 |

**Comparison with full model finetuning.** To establish an upper bound for our test-time adaptation, we also perform full-model finetuning (FT) on each input sequence. We test three variants: finetuning only the encoder (FT-Encoder), only the decoder head (FT-Head), or the entire model (FT-All). As shown in Tab. 7, FT-All achieves the highest performance, as expected. However, it only marginally surpasses CAPA while updating $\approx 2500\times$ more parameters. Interestingly, FT-Head performs worst, despite updating $80\times$ more parameters than our PEFT approach.

## 5 Conclusion

We have introduced CAPA, a new paradigm where depth completion is reframed as the parameter-efficient adaptation of the 3D foundation models at test time. This model-agnostic approach achieves state-of-the-art performance by grounding strong geometric priors using sparse, scene-specific gradients. While CAPA is significantly more efficient than other test-time optimization baselines, it remains slower than a single

Table 6: **Batch size vs. optimization steps** [AbsRel (%)↓].

|  | frames [%] | # steps | ScanNet | | | 7-Scenes | | Metropolis | |
|---|---|---|---|---|---|---|---|---|---|
|  |  |  | SIFT | 100 | <3m | SfM | 100 | 8-line | 16-line |
| CAPA_LoRA | 40 | 25 | 1.2 | 1.0 | 1.3 | 3.9 | 1.2 | 8.0 | 7.4 |
|  | 20 | 50 | 1.1 | 0.9 | 1.1 | 3.6 | 1.2 | 7.7 | 7.3 |
|  | 10 | 100 | 1.1 | 0.9 | 1.4 | 3.3 | 1.1 | 7.7 | 7.2 |
| CAPA_VPT | 40 | 25 | 1.3 | 1.1 | 1.4 | 3.2 | 1.4 | 8.5 | 8.1 |
|  | 20 | 50 | 1.3 | 1.0 | 1.5 | 3.1 | 1.3 | 8.3 | 7.8 |
|  | 10 | 100 | 1.2 | 1.0 | 1.4 | 2.7 | 1.2 | 8.2 | 7.7 |

Table 7: **Comparison to full-model finetuning** [AbsRel (%)↓].

|  | param. [%] | ScanNet | | | 7-Scenes | | Metropolis | |
|---|---|---|---|---|---|---|---|---|
|  |  | SIFT | 100 | <3m | SfM | 100 | 8-line | 16-line |
| VGGT | - | 2.5 | 2.1 | 2.1 | 6.0 | 5.3 | 10.7 | 10.9 |
| FT-All | 100.00 | 1.0 | 0.9 | 1.3 | 3.6 | 1.1 | 7.4 | OOM |
| FT-Encoder | 32.32 | 1.0 | 0.9 | 1.2 | 3.8 | 1.1 | 7.4 | 6.8 |
| FT-Head | 3.47 | 1.2 | 1.1 | 1.3 | 4.0 | 1.5 | 9.1 | 8.5 |
| LoRA-Encoder | 0.04 | 1.1 | 0.9 | 1.4 | 3.3 | 1.1 | 7.7 | 7.2 |
| VPT-Encoder | 0.04 | 1.2 | 1.0 | 1.4 | 2.7 | 1.2 | 8.2 | 7.7 |

forward pass. Future work will explore methods like trained optimizers to accelerate convergence. Another limitation is that CAPA cannot improve the input depth maps in regions not covered by sparse depth cues, and may fail to recover the correct geometry in cases where alreadt the frozen base model fails. We discuss representative failure cases in Appendix Sec. C.10.

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

# Appendix

# A    Implementation Details

## A.1    Hyper-parameters and Practical Settings

**Default settings.**    We employ AdamW optimizer with the default setting (*i.e.*, $\beta_1 = 0.9, \beta_2 = 0.999$). Additionally, we apply gradient clipping with a maximum norm of 1.0 to ensure training stability.    In our default setting, we use 100 optimization steps, with 10% frames at each step for multi-image input. Specifically, for UniDepthV2+CAPA$_{\text{VPT}}$, we use 20% frames, as the improvement from 10% is noticeable. The LoRA scaling factor is set to $\alpha = 2 \times r$ across all configurations. We summarize the LoRA rank, VPT prompt length, and learning rates (lr) for different base models in Tab. S1. For MoGe-2 and UniDepthV2 with the VPT variant, we apply a linear decay schedule that reduces the learning rate to $0.33\times$ its initial value.

Table S1: **Hyperparameters for different base models.**

| Base Model | LoRA Rank ($r$) | VPT Length ($t$) | LoRA lr | VPT lr |
|---|---|---|---|---|
| VGGT | 4 | 16 | $1 \times 10^{-3}$ | $2 \times 10^{-4}$ |
| MoGe-2 | 16 | 64 | $3 \times 10^{-4}$ | $2 \times 10^{-4}$ |
| UniDepthV2 | 16 | 96 | $3 \times 10^{-4}$ | $2 \times 10^{-4}$ |

**LoRA *vs*. VPT.**    In ablation studies, we found that both LoRA and VPT achieve comparable accuracy and are robust to hyperparameter changes. In practice, VPT performs slightly better when using generic prompts pre-tuned once on a few samples from an auxiliary dataset (see Sec. C.6). These learned prompts serve as a robust initialization that can be applied directly to new scenes, whereas LoRA typically starts with zero-initialized adapters.

**Insertion position.**    We found in practice that the performance is robust to which layers PEFT parameters are inserted into. For example, on the ScanNet dataset, applying LoRA to all 24 layers or only the deep 12 layers yields an identical AbsRel error of 0.011.  Restricting adaptation to the shallow 12 layers results in a marginal degradation to 0.012. We therefore default to inserting parameters into all layers for simplicity and maximizing model capacity.

## A.2    Inference Precision

We execute the ViT backbone using bfloat16 precision for efficiency.  However, to maintain the precision of depth predictions, we explicitly cast to float32 for the depth head.  We find that this mixed-precision approach yields higher accuracy, despite slightly higher inference latency and memory usage.

# B    Evaluation Details

## B.1    Baselines

We evaluate all baseline methods using their official open-source implementations and public checkpoints. We adhere to their original protocols regarding input resizing and output interpolation. For VGGT (Wang et al., 2025a), we use the predictions from the depth head. We employ the *Depth-Anything-V2-Large* checkpoint for DepthAnythingV2 (Yang et al., 2024b), *unidepth-v2-vitl14* for UniDepthV2 (Piccinelli et al., 2025), and *Metric-Video-Depth-Anything-Large* for VideoDA (Chen et al., 2025).  Regarding PriorDA (Wang et al., 2025e) and OMNI-DC (Zuo et al., 2025), we use their *v1.1* version of the codebases and checkpoints as they offer improved performance over the initial releases. For TestPromptDC (Jeong et al., 2025), we limit the number of condition points to 75K via random sampling to satisfy GPU memory constraints.

## B.2 Condition Point Selection

We follow prior work Zuo et al. (2025); Zuo & Deng (2024); Wang et al. (2025e) and define the following condition point selection strategies:

- **SfM**: We utilize actual Structure-from-Motion (SfM) points for 7-Scenes dataset, which were pre-computed using COLMAP (Schonberger & Frahm, 2016; Brachmann et al., 2021) and contain realistic noise. We filter these points by retaining only those below the 75th percentile for the bundle adjustment error. To further exclude outliers from reflective surfaces where SfM and sensor measurements are inconsistent, we filter against ground truth using the following consistency thresholds: AbsRel $< 0.1$, AbsErr $< 0.1$, and 3D point distance $< 0.1$.

- **SIFT**: As an alternative to SfM, we extract SIFT feature points from each image using OpenCV. These points are then filtered by the ground truth mask to ensure they correspond to valid depth measurements.

- **Random**: We randomly sample 100 pixels from regions with valid ground truth depth.

- **Limited Range**: We select all points with valid depth values within a predefined range.

- **LiDAR**: We simulate LiDAR patterns by uniformly sampling $N_L$ (corresponding to $N_L$-line) pitch angles within ground truth range, which is determined by the mean of pitch angle ranges of each image. For each sampled angle, we densely sample pixels horizontally to simulate LiDAR scan lines. This resulting mask is intersected with the ground truth mask to ensure that all selected points have valid depth values.

To ensure robustness, if the number of selected condition points is fewer than 5, we supplement them by randomly sampling additional points from the ground truth.

## B.3 Noise in Condition

We follow the noise injection strategy in OMNI-DC Zuo et al. (2025) to perturb the sampled ground truth depth values. Specifically, 10% of the condition points are randomly selected for corruption. For this subset, the original depth values are corrupted by adding noise drawn from a uniform distribution bounded by the 10th and 90th percentiles of the image's overall ground truth depth range. Note that for 7-Scenes SfM setting, we do not inject this artificial noise.

## B.4 Definitions of Evaluation Metrics

Let $\Omega$ denote the set of valid (dense) ground-truth pixels, $d_i$ the metric ground-truth depth at pixel $i$, and $\hat{d}_i$ the corresponding metric prediction.

The absolute relative error (AbsRel) is defined as

$$\text{AbsRel}(\hat{d}, d) = \frac{1}{|\Omega|} \sum_{i \in \Omega} \frac{|\hat{d}_i - d_i|}{d_i}. \tag{5}$$

The mean absolute error is defined as

$$\text{MAE}(\hat{d}, d) = \frac{1}{|\Omega|} \sum_{i \in \Omega} |\hat{d}_i - d_i|. \tag{6}$$

The root mean squared error is defined as

$$\text{RMSE}(\hat{d}, d) = \sqrt{\frac{1}{|\Omega|} \sum_{i \in \Omega} (\hat{d}_i - d_i)^2}. \tag{7}$$

For temporal consistency, we use the optical-flow-based warping error (OPW), following prior video-depth evaluation protocols (Wang et al., 2022; Ke et al., 2025a). Let $\hat{\mathbf{D}}_t$ be the predicted depth map for frame $\mathbf{I}_t$, and let $\Omega_{t+1}$ denote the evaluated pixels in frame $t+1$. Given the backward optical flow $\mathbf{F}_{t+1\Rightarrow t}$, we warp frame $t$ into the image coordinate system of frame $t+1$:

$$\widetilde{\mathbf{D}}_{t\rightarrow t+1} = \mathcal{W}(\hat{\mathbf{D}}_t, \mathbf{F}_{t+1\Rightarrow t}), \qquad \widetilde{\mathbf{I}}_{t\rightarrow t+1} = \mathcal{W}(\mathbf{I}_t, \mathbf{F}_{t+1\Rightarrow t}), \tag{8}$$

where $\mathcal{W}$ denotes bilinear warping. The pairwise OPW is defined as

$$\text{OPW}_t = \frac{1}{|\Omega_{t+1}|} \sum_{u \in \Omega_{t+1}} w_{t+1\Rightarrow t}^{(u)} \left\| \hat{\mathbf{D}}_{t+1}^{(u)} - \widetilde{\mathbf{D}}_{t\rightarrow t+1}^{(u)} \right\|. \tag{9}$$

The visibility weight is computed from the RGB warping discrepancy:

$$w_{t+1\Rightarrow t}^{(u)} = \exp\left( -\beta \left\| \mathbf{I}_{t+1}^{(u)} - \widetilde{\mathbf{I}}_{t\rightarrow t+1}^{(u)} \right\|_2^2 \right), \tag{10}$$

with $\beta = 50$. We set $w_{t+1\Rightarrow t}^{(u)} = 0$ for pixels without a valid backward-flow correspondence or with invalid warped depth. For a clip with $T$ frames, the clip-level OPW is obtained by averaging over consecutive frame pairs:

$$\text{OPW}_{\text{clip}} = \frac{1}{T-1} \sum_{t=1}^{T-1} \text{OPW}_t. \tag{11}$$

Unless stated otherwise, the values as per the above equations are multiplied by 100 for readability, for all reported metrics.

## C  Experimental Results

### C.1  Mean Absolute Error

In Tab. S2 and Tab. S3, we report MAE and RMSE for the quantitative comparison, as a supplement to AbsRel reported in the main table.

### C.2  Results under Noise-free Conditions

We also evaluate and compare the results under noise-free conditions (*i.e.*, utilizing exact ground truth values as conditions). Accuracy metrics (AbsRel and MAE) are reported in Tab. S4, and temporal consistency metric OPW is reported in Tab. S5.

### C.3  Additional Evaluation Datasets

To demonstrate the broad applicability of our method, we provide additional evaluations on NYU (Nathan Silberman & Fergus, 2012) (test split), KITTI (Geiger et al., 2013) (Eigen test), VOID (Wong et al., 2020) (test set), and DDAD (Guizilini et al., 2020) (validation split) . For NYU, we evaluate at the original image resolution, rather than the downsampled resolution used in some prior work, and use two sparsity patterns: SIFT keypoints and 100 randomly sampled valid ground-truth depth pixels. For KITTI, we simulate automotive LiDAR scan-line patterns with 8 and 16 lines. Following the main protocol, we corrupt 10% of the condition points with noise for NYU and KITTI. For VOID and DDAD, we use the condition points provided with the dataset: 150 and 500 points for VOID, and the LiDAR sparse depth for DDAD. As shown in Tab. S6, both variants of CAPA generalize well to these domains across diverse conditioning patterns.

Table S2: **MAE**(%)↓ in quantitative comparison of CAPA with baseline methods.

| Dataset | ScanNet | | | 7-Scenes | | | iBims | | | Metropolis | | |
|---|---|---|---|---|---|---|---|---|---|---|---|---|
| Condition | SIFT | 100 | $< 3m$ | SfM | 100 | $< 3m$ | SIFT | 100 | $< 5m$ | 8-line | 16-line | 32-line |
| DepthAnythingV2 | 9.2 | 7.5 | 11.8 | 9.9 | 8.3 | 8.7 | 13.0 | 11.9 | 75.8 | 3569.9 | 3670.8 | 3682.4 |
| UniDepthV2 | 6.1 | 5.3 | 5.9 | 7.6 | 6.7 | 6.7 | 10.6 | 9.1 | 16.0 | 1401.8 | 1316.9 | 1305.3 |
| MoGe-2 | 7.7 | 6.7 | 7.2 | 7.0 | 6.0 | 6.1 | 10.0 | 8.6 | 15.8 | 1307.9 | 1253.7 | 1244.6 |
| VGGT | 4.8 | 4.4 | 4.5 | 8.9 | 8.2 | 8.3 | 14.1 | 12.4 | 20.4 | 861.1 | 839.7 | 833.3 |
| VideoDA | 9.2 | 7.8 | 8.5 | 10.8 | 9.2 | 9.3 | 16.5 | 13.8 | 21.3 | 2341.8 | 2219.1 | 2199.5 |
| PromptDA | 27.6 | 22.7 | 23.8 | 15.0 | 12.5 | 8.8 | 34.7 | 26.1 | 49.6 | 1597.2 | 1108.9 | 963.8 |
| PriorDA-v1.1 | 5.8 | 4.7 | 26.4 | 6.6 | 5.6 | 21.0 | 9.0 | 8.8 | 72.8 | 1352.5 | 1103.9 | 978.7 |
| OMNI-DC-v1.1 | 3.7 | 3.0 | 14.8 | 6.0 | 4.1 | 11.0 | 5.3 | **5.3** | 62.1 | 976.7 | 718.1 | 590.9 |
| Marigold-DC | 7.7 | 7.1 | 7.7 | 7.4 | 7.3 | 4.0 | 22.8 | 20.2 | 38.6 | 1389.0 | 945.1 | 710.5 |
| TestPromptDC | 7.7 | 7.1 | 6.5 | 5.7 | 7.2 | 5.2 | 18.4 | 16.1 | 29.1 | 906.8 | 665.9 | 546.5 |
| CAPA$_{LoRA}$ | **2.0** | **1.8** | 3.1 | 5.2 | **1.9** | **2.0** | 5.4 | 7.3 | 15.7 | **456.8** | **401.5** | **375.1** |
| CAPA$_{LoRA}$ | 2.2 | 1.9 | **2.7** | **5.0** | 2.1 | 2.2 | **4.6** | 6.1 | **15.4** | 482.3 | 434.2 | 417.4 |
| CAPA$_{LoRA}$ + MoGe-2 | 2.6 | 2.1 | 3.6 | 5.3 | 2.6 | 2.9 | 6.3 | 7.5 | 15.5 | 596.8 | 483.5 | 447.2 |
| CAPA$_{VPT}$ + MoGe-2 | 2.9 | 2.4 | 4.0 | 5.4 | 3.1 | 3.5 | 5.0 | 7.0 | 17.5 | 640.1 | 537.4 | 508.1 |
| CAPA$_{LoRA}$ + UniDepthV2 | 2.6 | 2.2 | 3.8 | 5.5 | 2.8 | 3.0 | 7.0 | 8.5 | 19.2 | 594.1 | 480.8 | 447.3 |
| CAPA$_{VPT}$ + UniDepthV2 | 2.6 | 2.2 | 4.3 | 5.6 | 2.7 | 3.3 | 6.8 | 8.1 | 21.3 | 611.8 | 510.1 | 475.7 |

Table S3: **RMSE**(%)↓ in quantitative comparison of CAPA with baseline methods.

| Dataset | ScanNet | | | 7-Scenes | | | iBims | | | Metropolis | | |
|---|---|---|---|---|---|---|---|---|---|---|---|---|
| Condition | SIFT | 100 | $< 3m$ | SfM | 100 | $< 3m$ | SIFT | 100 | $< 5m$ | 8-line | 16-line | 32-line |
| DepthAnythingV2 | 70.1 | 18.8 | 317.6 | 18.4 | 16.0 | 16.2 | 24.4 | 25.0 | 9905.9 | 317854.6 | 363883.5 | 339026.5 |
| UniDepthV2 | 13.0 | 12.3 | 12.9 | 15.6 | 15.0 | 15.0 | 22.6 | 21.8 | 30.2 | 2849.4 | 2845.6 | 2824.4 |
| MoGe-2 | 15.3 | 14.3 | 15.0 | 14.5 | 13.8 | 13.8 | 20.5 | 19.7 | **28.9** | 2424.5 | 2372.7 | 2370.3 |
| VGGT | 11.6 | 11.3 | 11.5 | 19.6 | 19.2 | 19.3 | 30.2 | 29.7 | 39.5 | 2079.8 | 2045.5 | 2048.1 |
| VideoDA | 15.4 | 14.1 | 15.0 | 18.5 | 16.7 | 16.6 | 29.8 | 28.1 | 37.4 | 3758.6 | 3710.5 | 3667.5 |
| PromptDA | 36.7 | 30.0 | 33.3 | 24.1 | 19.5 | 16.9 | 51.7 | 42.8 | 82.9 | 2869.2 | 2167.4 | 1943.3 |
| PriorDA-v1.1 | 11.6 | 10.9 | 39.7 | 13.1 | 12.8 | 29.9 | 19.2 | 20.7 | 105.2 | 2370.4 | 2059.7 | 1924.4 |
| OMNI-DC-v1.1 | 9.0 | 8.9 | 31.5 | 12.6 | 11.6 | 24.7 | **15.3** | 16.6 | 102.5 | 2002.1 | 1580.7 | 1403.1 |
| Marigold-DC | 16.1 | 17.4 | 18.4 | 14.7 | 16.9 | 11.0 | 44.3 | 48.0 | 70.6 | 2543.1 | 1852.8 | 1406.4 |
| TestPromptDC | 16.0 | 17.2 | 12.3 | 12.0 | 16.8 | 9.7 | 37.7 | 38.7 | 49.9 | 1776.3 | 1357.8 | **1116.6** |
| CAPA$_{LoRA}$ | **5.3** | **5.3** | 9.8 | 11.9 | **6.1** | **6.6** | 18.1 | 25.2 | 34.8 | **1307.2** | **1200.4** | 1147.1 |
| CAPA$_{LoRA}$ | 5.7 | 5.5 | **8.9** | **11.1** | 6.3 | 7.0 | 16.2 | 20.8 | 34.6 | 1348.1 | 1259.2 | 1232.1 |
| CAPA$_{LoRA}$ + MoGe-2 | 6.4 | 6.4 | 10.0 | 11.3 | 7.6 | 8.7 | 17.0 | 23.2 | 31.8 | 1499.6 | 1325.6 | 1260.5 |
| CAPA$_{VPT}$ + MoGe-2 | 6.9 | 6.8 | 11.1 | 11.6 | 9.2 | 10.2 | **15.3** | 22.7 | 37.7 | 1567.1 | 1401.5 | 1356.3 |
| CAPA$_{LoRA}$ + UniDepthV2 | 6.8 | 6.8 | 10.4 | 12.3 | 8.2 | 8.9 | 18.4 | 25.2 | 38.4 | 1494.1 | 1322.1 | 1255.6 |
| CAPA$_{VPT}$ + UniDepthV2 | 6.5 | 6.5 | 11.8 | 12.3 | 8.1 | 9.7 | 17.5 | 24.6 | 44.2 | 1529.1 | 1376.2 | 1310.7 |

## C.4 Varying condition point density

While the main paper focuses on sparse or incomplete settings, we extend our analysis here to varying point densities. Specifically, we evaluate performance on the ablation subset of ScanNet by randomly sampling condition points at various densities, with noise injection applied.

From Fig. S1 and Tab. S7 we can see that given denser condition points ($> 500$) , per-image optimization benefits more in terms of accuracy, while per-sequence tuning (with shared parameters) still maintains the optimal temporal consistency.

Table S4: Quantitative comparison **under noise-free conditions** (*i.e.*, using ground-truth). Numbers are presented in percentage (%).

| Dataset | ScanNet | | | | | | 7-Scenes | | | | | |
|---|---|---|---|---|---|---|---|---|---|---|---|---|
| Condition | SIFT | | 100 | | $< 3m$ | | SfM Masked Ground Truth | | 100 | | $< 3m$ | |
| | AbsRel↓ | MAE↓ | AbsRel↓ | MAE↓ | AbsRel↓ | MAE↓ | AbsRel↓ | MAE↓ | AbsRel↓ | MAE↓ | AbsRel↓ | MAE↓ |
| DepthAnythingV2 | 4.9 | 9.5 | 3.8 | 7.5 | 4.9 | 12.7 | 5.2 | 9.6 | 4.6 | 8.3 | 4.6 | 8.7 |
| UniDepthV2 | 3.2 | 6.0 | 2.7 | 5.3 | 2.8 | 5.9 | 4.0 | 7.3 | 3.7 | 6.6 | 3.6 | 6.7 |
| MoGe-2 | 4.0 | 7.6 | 3.3 | 6.6 | 3.3 | 7.2 | 3.6 | 6.5 | 3.3 | 6.0 | 3.3 | 6.0 |
| VGGT | 2.2 | 4.7 | 1.9 | 4.4 | 1.9 | 4.5 | 4.4 | 8.5 | 4.2 | 8.1 | 4.2 | 8.2 |
| VideoDA | 4.9 | 9.1 | 4.1 | 7.7 | 4.2 | 8.5 | 6.0 | 10.7 | 5.1 | 9.1 | 5.0 | 9.2 |
| PromptDA | 14.9 | 25.5 | 11.1 | 20.1 | 9.7 | 21.2 | 8.0 | 14.7 | 5.8 | 10.7 | 3.2 | 7.6 |
| PriorDA-v1.1 | 2.5 | 4.4 | 1.6 | 3.3 | 9.2 | 24.0 | 3.0 | 5.5 | 2.5 | 4.5 | 8.0 | 18.6 |
| OMNI-DC-v1.1 | 1.8 | 3.4 | 1.4 | 2.8 | 2.7 | 8.5 | 2.4 | 4.5 | 2.1 | 3.8 | 1.9 | 4.8 |
| Marigold-DC | 3.0 | 5.4 | 1.8 | 3.7 | 2.2 | 7.6 | 3.4 | 6.2 | 2.5 | 4.6 | 1.6 | 4.0 |
| TestPromptDC | 1.5 | 2.8 | 1.1 | 2.3 | **0.9** | 2.9 | 2.0 | 3.9 | 1.7 | 3.3 | **0.9** | 2.3 |
| CAPA$_{\text{LoRA}}$ | **1.0** | **2.0** | 0.9 | **1.8** | 1.0 | 2.9 | **1.3** | **2.5** | 1.0 | **1.8** | **0.9** | **2.0** |
| CAPA$_{\text{VPT}}$ | 1.1 | 2.2 | 0.9 | 1.9 | 1.0 | **2.7** | 1.4 | 2.7 | 1.1 | 2.0 | 1.0 | 2.2 |
| CAPA$_{\text{LoRA}}$ + MoGe-2 | 1.3 | 2.5 | 1.0 | 2.1 | 1.3 | 3.4 | 1.9 | 3.7 | 1.3 | 2.5 | 1.3 | 2.8 |
| CAPA$_{\text{VPT}}$ + MoGe-2 | 1.5 | 2.8 | 1.2 | 2.3 | 1.5 | 3.9 | 2.1 | 4.0 | 1.6 | 3.1 | 1.6 | 3.4 |
| CAPA$_{\text{LoRA}}$ + UniDepthV2 | 1.4 | 2.5 | 1.1 | 2.2 | 1.3 | 3.7 | 2.0 | 3.8 | 1.4 | 2.6 | 1.3 | 2.8 |
| CAPA$_{\text{VPT}}$ + UniDepthV2 | 1.4 | 2.5 | 1.1 | 2.1 | 1.4 | 4.1 | 2.1 | 4.1 | 1.4 | 2.6 | 1.5 | 3.2 |

| Dataset | iBims | | | | | | Metropolis | | | | | |
|---|---|---|---|---|---|---|---|---|---|---|---|---|
| Condition | SIFT | | 100 | | $< 5m$ | | 8-line | | 16-line | | 32-line | |
| | AbsRel↓ | MAE↓ | AbsRel↓ | MAE↓ | AbsRel↓ | MAE↓ | AbsRel↓ | MAE↓ | AbsRel↓ | MAE↓ | AbsRel↓ | MAE↓ |
| DepthAnythingV2 | 4.0 | 13.1 | 3.3 | 12.1 | 9.8 | 75.5 | 59.4 | 3964.5 | 56.7 | 3794.1 | 60.0 | 4018.0 |
| UniDepthV2 | 3.4 | 10.7 | 2.7 | 9.1 | 3.2 | 16.1 | 29.1 | 1385.3 | 28.7 | 1315.0 | 28.0 | 1301.8 |
| MoGe-2 | 3.3 | 9.9 | 2.6 | 8.5 | 3.1 | 15.9 | 21.5 | 1302.3 | 20.1 | 1250.4 | 19.8 | 1242.9 |
| VGGT | 4.0 | 14.1 | 3.3 | 12.3 | 3.8 | 20.3 | 10.1 | 858.2 | 10.2 | 839.4 | 9.9 | 832.5 |
| VideoDA | 5.0 | 16.0 | 3.9 | 13.6 | 4.5 | 20.9 | 49.2 | 2336.6 | 45.9 | 2221.2 | 44.9 | 2199.2 |
| PromptDA | 9.5 | 29.3 | 5.6 | 20.2 | 6.8 | 46.7 | 24.3 | 1597.2 | 16.1 | 1128.3 | 13.7 | 1014.5 |
| PriorDA-v1.1 | 2.1 | 6.8 | 1.7 | 6.0 | 13.5 | 67.5 | 21.1 | 1128.9 | 14.1 | 797.5 | 10.1 | 604.6 |
| OMNI-DC-v1.1 | 1.5 | 4.9 | 1.4 | 4.9 | 5.8 | 40.6 | 13.5 | 890.3 | 10.0 | 628.2 | 7.5 | 470.0 |
| Marigold-DC | 4.9 | 14.3 | 2.5 | 8.8 | 5.1 | 38.3 | 27.9 | 1267.6 | 18.8 | 807.5 | 13.7 | 571.2 |
| TestPromptDC | 2.1 | 6.6 | 1.6 | 5.7 | 2.5 | 19.9 | 13.0 | 743.9 | 8.9 | 481.9 | 6.8 | **354.5** |
| CAPA$_{\text{LoRA}}$ | **1.2** | **4.1** | 1.3 | 4.7 | 1.8 | 14.4 | **7.7** | **454.1** | **7.0** | **398.9** | **6.7** | 373.8 |
| CAPA$_{\text{VPT}}$ | 1.3 | 4.3 | 1.3 | **4.6** | 1.8 | **14.0** | 8.1 | 478.7 | 7.6 | 432.8 | 7.4 | 416.3 |
| CAPA$_{\text{LoRA}}$ + MoGe-2 | 1.2 | 4.0 | 1.2 | 4.2 | 1.9 | 14.2 | 10.0 | 601.6 | 8.4 | 482.5 | 7.9 | 445.7 |
| CAPA$_{\text{VPT}}$ + MoGe-2 | 1.3 | 4.2 | 1.2 | 4.5 | 2.1 | 15.9 | 10.9 | 638.6 | 9.6 | 533.9 | 9.1 | 505.9 |
| CAPA$_{\text{LoRA}}$ + UniDepthV2 | 1.3 | 4.4 | 1.3 | 4.7 | 2.6 | 19.6 | 9.8 | 585.8 | 8.2 | 481.6 | 7.9 | 447.0 |
| CAPA$_{\text{VPT}}$ + UniDepthV2 | 1.5 | 4.7 | 1.3 | 4.7 | 2.7 | 21.2 | 10.1 | 605.8 | 8.8 | 514.1 | 8.3 | 473.5 |

### C.5 Impact of LoRA Rank and VPT Token Length

We study the impact of the rank of LoRA and the token length of VPT, the results are reported in Tab. S8. CAPA is robust to parameter changes, maintaining performance with as few as 0.2M trainable parameters ($r = 2$ or $t = 8$).

### C.6 VPT Token Initialization.

We study different variants of token initialization for Visual Prompt Tuning. The results are reported in Tab. S9. We find that with random initialization like Xavier (Glorot & Bengio, 2010), CAPA already gives significant improvement over the base model. When initialized with pre-tuned tokens, the performance can be further boosted in most cases. Here we pre-tune these tokens on just a few ScanNet++ (Yeshwanth et al., 2023) sequences.

Table S5: Temporal consistency evaluation [OPW%↓] **under noise-free conditions**.

| Dataset | ScanNet | | | 7-Scenes | | | Metropolis | | |
|---|---|---|---|---|---|---|---|---|---|
| Condition | SIFT | 100 | $< 3m$ | SfM | 100 | $< 3m$ | 8-line | 16-line | 32-line |
| UniDepthV2 | 2.7 | 2.5 | 2.5 | 2.9 | 2.9 | 2.6 | 165.2 | 152.2 | 141.1 |
| MoGe-2 | 4.1 | 3.7 | 3.6 | 3.1 | 2.9 | 2.7 | 219.3 | 205.3 | 199.1 |
| VGGT | 2.3 | 2.2 | 2.0 | 3.1 | 3.2 | 2.9 | 149.4 | 152.5 | 149.0 |
| VideoDA | 3.3 | 3.2 | 2.8 | 3.3 | 3.6 | 2.9 | 179.3 | 160.4 | 141.5 |
| PriorDA-v1.1 | 5.0 | 4.7 | 3.7 | 3.5 | 4.9 | 4.1 | 268.9 | 233.2 | 195.0 |
| OMNI-DC-v1.1 | 3.2 | 3.0 | 9.2 | 3.3 | 3.7 | 10.0 | 171.2 | 174.5 | 171.3 |
| Marigold-DC | 8.1 | 9.0 | 2.6 | 4.8 | 8.9 | 3.0 | 257.7 | 213.3 | 174.7 |
| TestPromptDC | 8.9 | 9.3 | 2.3 | 3.7 | 9.1 | 2.5 | 185.6 | 153.9 | 138.4 |
| CAPA$_{\text{LoRA}}$ | **1.8** | **1.8** | **1.9** | **2.4** | **2.3** | **2.3** | **119.4** | **120.0** | **118.4** |
| CAPA$_{\text{VPT}}$ | **1.8** | **1.8** | **1.9** | **2.4** | **2.3** | **2.3** | 119.8 | **120.0** | 118.6 |
| CAPA$_{\text{LoRA}}$ + MoGe-2 | 2.1 | 2.0 | 2.0 | 2.5 | 2.5 | 2.4 | 120.9 | 118.3 | 116.3 |
| CAPA$_{\text{VPT}}$ + MoGe-2 | 2.2 | 2.1 | 2.1 | 2.5 | 2.5 | 2.5 | 121.9 | 119.9 | 118.4 |
| CAPA$_{\text{LoRA}}$ + UniDepthV2 | 2.1 | 2.0 | 2.0 | 2.5 | 2.5 | 2.4 | 119.8 | 120.9 | 119.3 |
| CAPA$_{\text{VPT}}$ + UniDepthV2 | 2.1 | 2.0 | 2.0 | 2.5 | 2.6 | 2.4 | 122.6 | 119.9 | 118.9 |

Table S6: **Quantitative evaluation on additional datasets** [AbsRel%↓, MAE%↓].

| | NYU | | | | KITTI | | | | VOID | | | | DDAD | |
|---|---|---|---|---|---|---|---|---|---|---|---|---|---|---|
| | SIFT | | 100 | | 8-line | | 16-line | | 150 | | 500 | | lidar | |
| | AbsRel | MAE | AbsRel | MAE | AbsRel | MAE | AbsRel | MAE | AbsRel | MAE | AbsRel | MAE | AbsRel | MAE |
| VGGT | 3.5 | 9.6 | 3.2 | 9.0 | 6.5 | 139.5 | 6.4 | 139.7 | 5.7 | 5.1 | 5.4 | 4.5 | 13.5 | 550.7 |
| PriorDA-v1.1 | 2.8 | 7.1 | 3.0 | 7.7 | 5.6 | 112.4 | 5.9 | 111.2 | 4.4 | 4.7 | 3.6 | 3.9 | 4.7 | 219.8 |
| OMNI-DC-v1.1 | 1.5 | 4.1 | **1.8** | **5.0** | 3.6 | 81.0 | 4.1 | 98.6 | 3.8 | 4.4 | 3.1 | 3.7 | **2.8** | **100.1** |
| Marigold-DC | 3.6 | 9.2 | 5.8 | 14.5 | 9.1 | 145.5 | 7.7 | 126.4 | 5.3 | 5.2 | 4.4 | 4.3 | 10.6 | 297.6 |
| TestPromptDC | 4.8 | 11.0 | 5.2 | 13.0 | 8.5 | 113.9 | 7.5 | 103.3 | 3.7 | 4.2 | 3.1 | 3.4 | 4.6 | 156.2 |
| CAPA$_{\text{LoRA}}$ | **1.3** | **3.6** | 2.2 | 6.3 | **2.3** | **57.7** | **2.0** | **50.2** | **3.2** | **4.0** | **2.6** | **3.2** | 3.2 | 129.8 |
| CAPA$_{\text{PT}}$ | **1.3** | **3.6** | 1.9 | 5.6 | 2.4 | 58.1 | 2.1 | 51.1 | **3.2** | **4.0** | 2.7 | 3.3 | 3.4 | 141.0 |

## C.7 Parameter Sharing for MoGe-2

We further validate the generalizability of our parameter sharing method by applying it to MoGe-2 (Wang et al., 2025c), a single-image depth estimation model. The results in Tab. S10 demonstrate that this technique is not limited to multi-view architectures; it significantly improves both the accuracy and temporal consistency of the single-image base model.

## C.8 Efficiency Analysis

Table S7: **Conditioned on various densities**. Numbers are presented in percentage (%).

| Num. Sampled Points | 50 | | 100 | | 500 | | 1000 | | 1500 | | 2000 | | 2500 | |
|---|---|---|---|---|---|---|---|---|---|---|---|---|---|---|
| | AbsRel↓ | OPW↓ | AbsRel↓ | OPW↓ | AbsRel↓ | OPW↓ | AbsRel↓ | OPW↓ | AbsRel↓ | OPW↓ | AbsRel↓ | OPW↓ | AbsRel↓ | OPW↓ |
| VGGT | 2.1 | 2.6 | 2.1 | 2.4 | 2.1 | 2.3 | 2.1 | 2.3 | 2.1 | 2.2 | 2.1 | 2.2 | 2.1 | 2.2 |
| PriorDA-v1.1 | 3.1 | 6.0 | 2.5 | 4.9 | 1.8 | 3.5 | 1.7 | 3.3 | 1.7 | 3.1 | 1.7 | 3.1 | 1.7 | 3.1 |
| OMNI-DC-v1.1 | 1.9 | 3.7 | 1.5 | 3.1 | 1.0 | 2.5 | 0.8 | 2.3 | 0.8 | 2.3 | **0.7** | 2.3 | 0.7 | 2.2 |
| Marigold-DC | 5.0 | 12.4 | 3.7 | 9.7 | 1.8 | 4.6 | 1.4 | 3.6 | 1.3 | 3.2 | 1.2 | 3.1 | 1.1 | 2.9 |
| TestPromptDC | 4.4 | 11.1 | 4.0 | 10.0 | 3.1 | 6.3 | 2.9 | 5.0 | 2.8 | 4.3 | 2.8 | 3.9 | 2.7 | 3.6 |
| CAPA$_{\text{LoRA}}$ (per-image) | 2.4 | 6.5 | 1.7 | 4.9 | 0.9 | 2.8 | **0.7** | 2.4 | **0.7** | 2.3 | **0.7** | 2.2 | **0.6** | 2.2 |
| CAPA$_{\text{LoRA}}$ (per-sequence) | **1.0** | **2.0** | 0.9 | 1.9 | 0.8 | 1.9 | 0.8 | 1.9 | 0.8 | 1.8 | 0.8 | 1.8 | 0.8 | **1.8** |
| CAPA$_{\text{VPT}}$ (per-image) | 2.0 | 5.5 | 1.5 | 4.1 | 0.9 | 2.6 | 0.8 | 2.3 | **0.7** | 2.2 | **0.7** | 2.1 | 0.7 | 2.1 |
| CAPA$_{\text{VPT}}$ (per-sequence) | 1.1 | **2.0** | 1.0 | 1.9 | 0.9 | 1.9 | 0.9 | 1.9 | 0.9 | 1.9 | 0.9 | 1.9 | 0.9 | 1.9 |

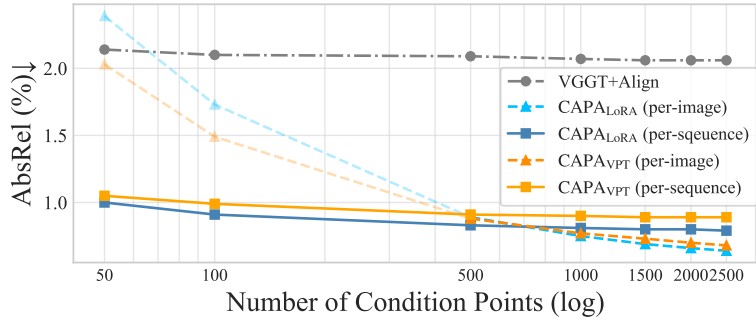

Figure S1: **Per-frame tuning vs. per-sequence tuning** under various condition point density. When the condition is not very sparse ($> 500$), per-image tuning is preferred.

Table S8: **Ablation of LoRA rank and VPT token length** [AbsRel%↓].

| Method | Rank / Token Len. | #Param. | ScanNet | | | 7-Scenes | | Metropolis | |
|---|---|---|---|---|---|---|---|---|---|
| | | | SIFT | 100 | <3m | SfM | 100 | 8-line | 16-line |
| CAPA**LoRA** | $r = 2$ | 0.20M | 1.10 | 0.92 | 1.42 | 3.88 | 1.11 | 7.84 | 7.24 |
| | $r = 4$ | 0.39M | 1.08 | 0.91 | 1.45 | 3.27 | 1.12 | 7.72 | 7.17 |
| | $r = 8$ | 0.79M | 1.07 | 0.90 | 1.22 | 3.52 | 1.11 | 7.76 | 7.02 |
| | $r = 16$ | 1.57M | 1.10 | 0.95 | 1.38 | 2.90 | 1.09 | 7.84 | 7.00 |
| CAPA**VPT** | $t = 8$ | 0.20M | 1.21 | 1.00 | 1.47 | 3.07 | 1.24 | 8.46 | 7.85 |
| | $t = 16$ | 0.39M | 1.18 | 0.98 | 1.46 | 3.35 | 1.26 | 8.59 | 7.99 |
| | $t = 32$ | 0.79M | 1.18 | 1.02 | 1.42 | 2.80 | 1.27 | 11.16 | 7.98 |
| | $t = 64$ | 1.57M | 1.32 | 1.07 | 1.46 | 4.07 | 1.31 | 11.40 | 9.44 |

**Runtime details.** The end-to-end runtime comparison is reported in the main paper in Tab. 5. In this context, we note that the added cost of CAPA mainly comes from PEFT optimization rather than affine alignment or final inference. Affine alignment only estimates two values (global scale and shift) from sparse correspondences, so it is negligible compared with the forward/backward passes required during adaptation. After adaptation, the final inference pass has the same cost as the frozen base model.

**Memory efficiency.** Unlike full model fine-tuning (FT-All), which requires storing gradients and optimizer states for the entire backbone, CAPA minimizes the memory (VRAM) footprint during optimization by adopting PEFT techniques, which update only 0.39M parameters ($\approx 0.04\%$ of the model), reducing peak VRAM usage. This allows for the adaptation of large-scale foundation models on standard consumer-grade GPUs, where full fine-tuning may result in Out-Of-Memory (OOM).

## C.9 Application to Generative Novel View Synthesis

We demonstrate the practical impact of our refined depth maps on downstream novel view synthesis using the Gen3C model (Ren et al., 2025), as illustrated in Fig. S2. By default, Gen3C relies on standard MoGe-2 depth estimates (indicated by green arrows), which may produce noticeable synthesis artifacts (indicated

Table S9: **Ablation of VPT token initialization**.

| Token Initialization | ScanNet | | | 7-Scenes | | Metropolis | |
|---|---|---|---|---|---|---|---|
| | SIFT | 100 | <3m | SfM | 100 | 8-line | 16-line |
| Original + Align | 2.5 | 2.1 | 2.1 | 6.0 | 5.3 | 10.7 | 10.9 |
| Xavier | 1.2 | 1.0 | 1.5 | 3.3 | 1.3 | 8.6 | 8.0 |
| Pre-tuned | 1.2 | 1.0 | 1.4 | 2.7 | 1.2 | 8.2 | 7.7 |

Table S10: **Effect of parameter sharing across video frames**, taking MoGe-2 as base model.

| Method | Parameter Sharing | ScanNet (100) | | 7-Scenes (SfM) | | Metropolis (8-line) | |
|---|---|---|---|---|---|---|---|
| | | AbsRel↓ | OPW↓ | AbsRel↓ | OPW↓ | AbsRel↓ | OPW↓ |
| MoGe-2 + | Frame | 1.9 | 5.0 | 3.5 | 3.6 | 11.0 | 164.7 |
| CAPA$_{\text{LoRA}}$ | Sequence | 1.1 | 2.1 | 3.1 | 2.6 | 9.9 | 118.2 |
| MoGe-2 + | Frame | 1.8 | 4.8 | 3.5 | 3.7 | 11.4 | 168.4 |
| CAPA$_{\text{VPT}}$ | Sequence | 1.2 | 2.2 | 3.2 | 2.7 | 11.0 | 118.6 |

by red arrows). Replacing this input with our depth predictions, conditioned on sparse LiDAR, effectively corrects the geometry, resulting in synthesized views that are geometrically consistent and largely free of artifacts.

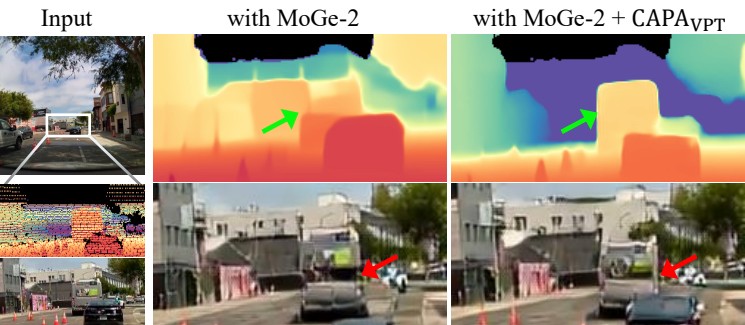

Figure S2: **Novel view synthesis results** using Gen3C, with depth map from Moge-2 and CAPA$_{\text{VPT}}$. The viewpoint is moved by 1m higher than the input.

## C.10  Failure Cases

We note that CAPA relies on the geometric prior of the pre-trained base model. Consequently, in extreme scenarios where the foundation model fails completely, such as on visual illusions or complex non-Lambertian reflective surfaces (see Fig. S3), the adaptation may be insufficient to fully correct the geometry using only sparse cues, potentially resulting in noisy or blurry predictions. Furthermore, challenges remain when the physical definition of depth differs between the visual prior (typically surface depth) and the sparse measurements (*e.g.*, LiDAR penetrating glass or measuring reflected distance).

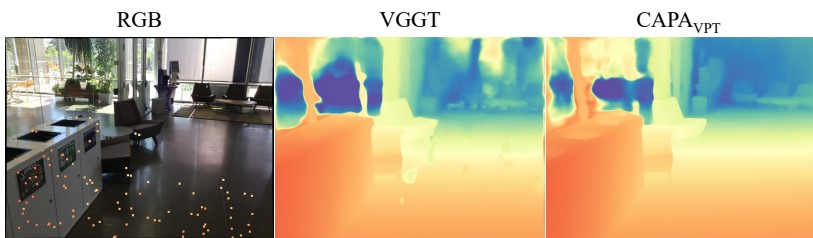

Figure S3: **Failure case,** where the base model fails on mirrors and CAPA cannot fully recover due to insufficient condition points.

## C.11  Additional Qualitative Results

Additional qualitative comparisons on the test datasets are shown in Fig. S5, including error maps, depth maps, and point clouds. Interactive comparisons can be found on the attached website. These extended

results provide a broader view of CAPA's ability to recover consistent global geometry and fine structural details across diverse indoor and outdoor scenes.

To further investigate model robustness, Fig. S4 showcases an example that highlights the specific challenge of handling noisy condition points.

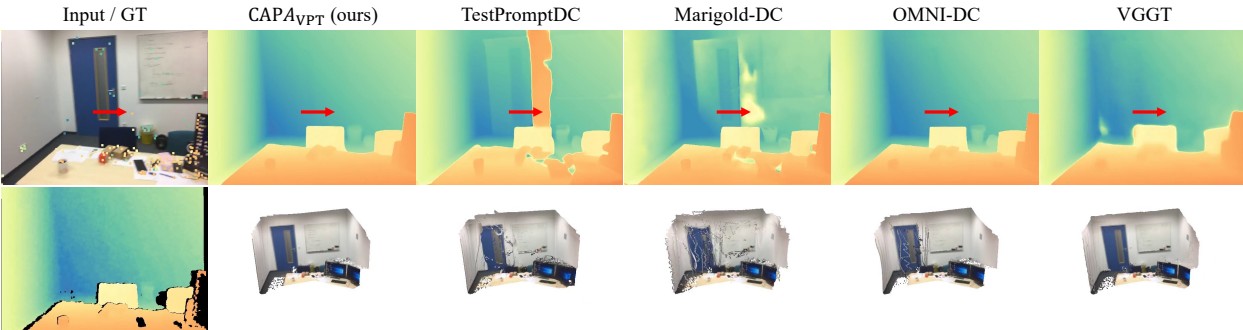

Figure S4: **Robustness to noisy condition points.** By injecting prompts directly into pixel space, TestPromptDC is highly susceptible to hallucinating structures around noisy inputs, *e.g.*, the point indicated by the red arrow. Marigold-DC also exhibits minor artifacts near these anomalies. In contrast, CAPA filters out this noise by leveraging the frozen global geometric prior of the base model, yielding clean reconstructions.

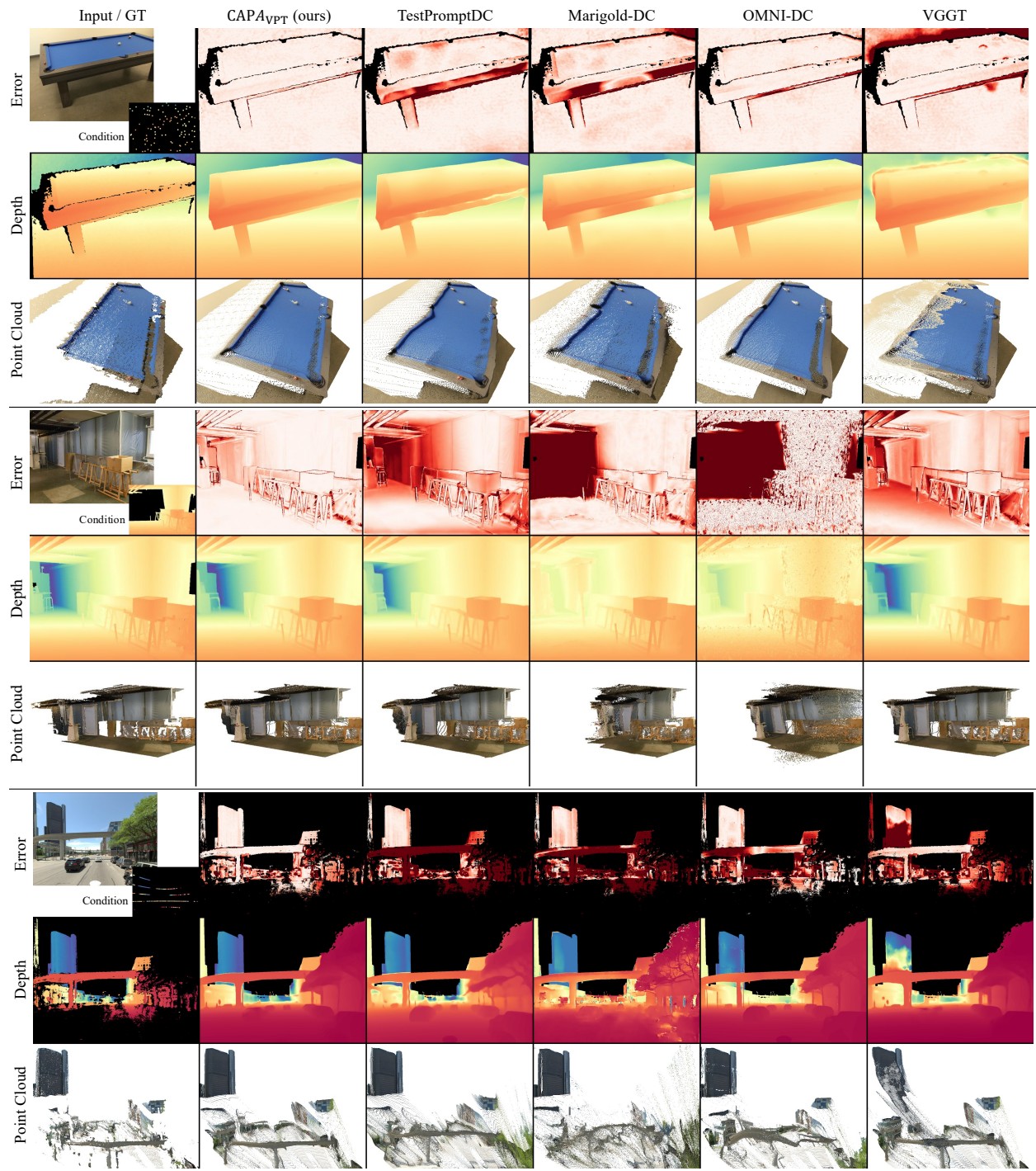

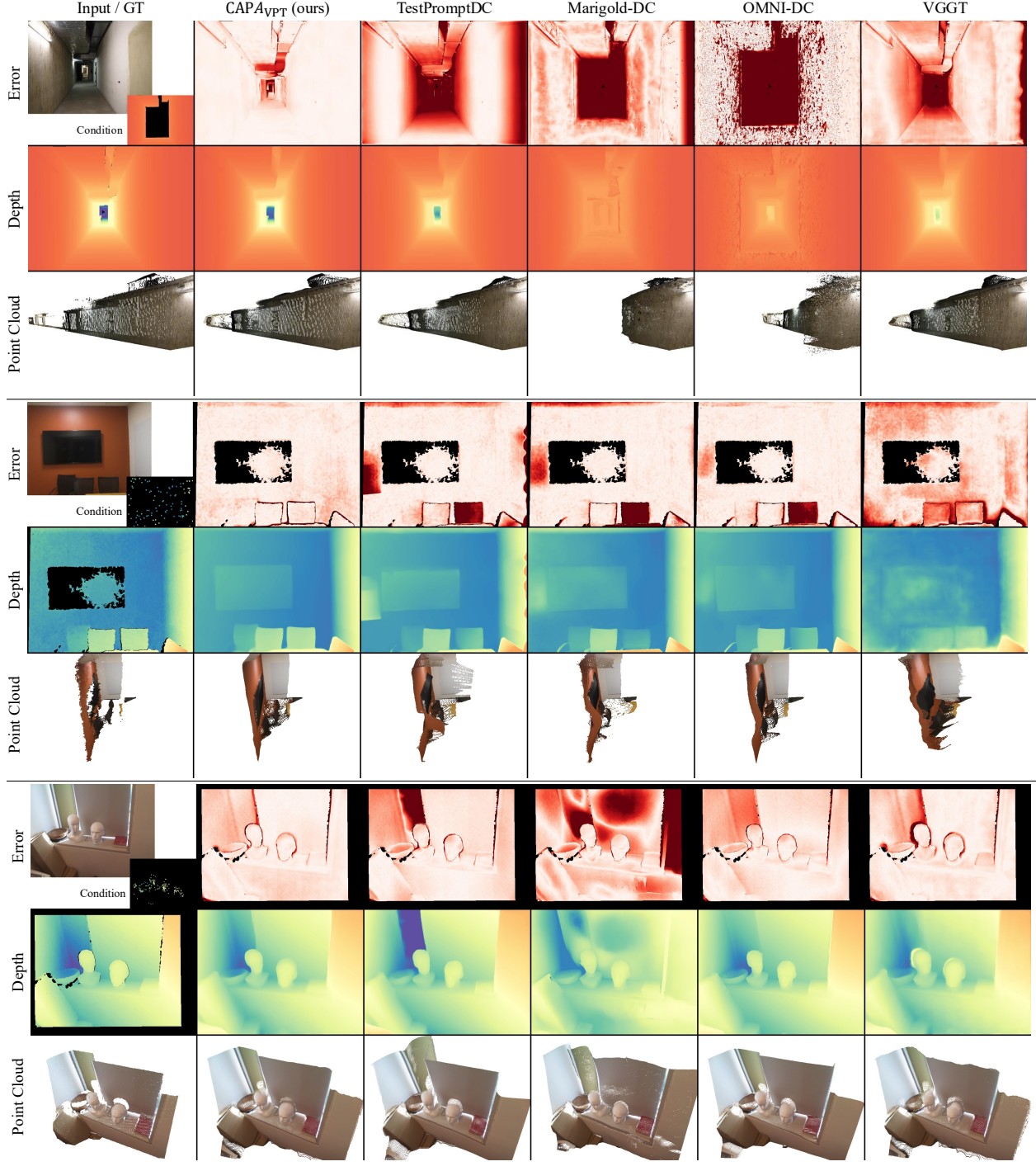

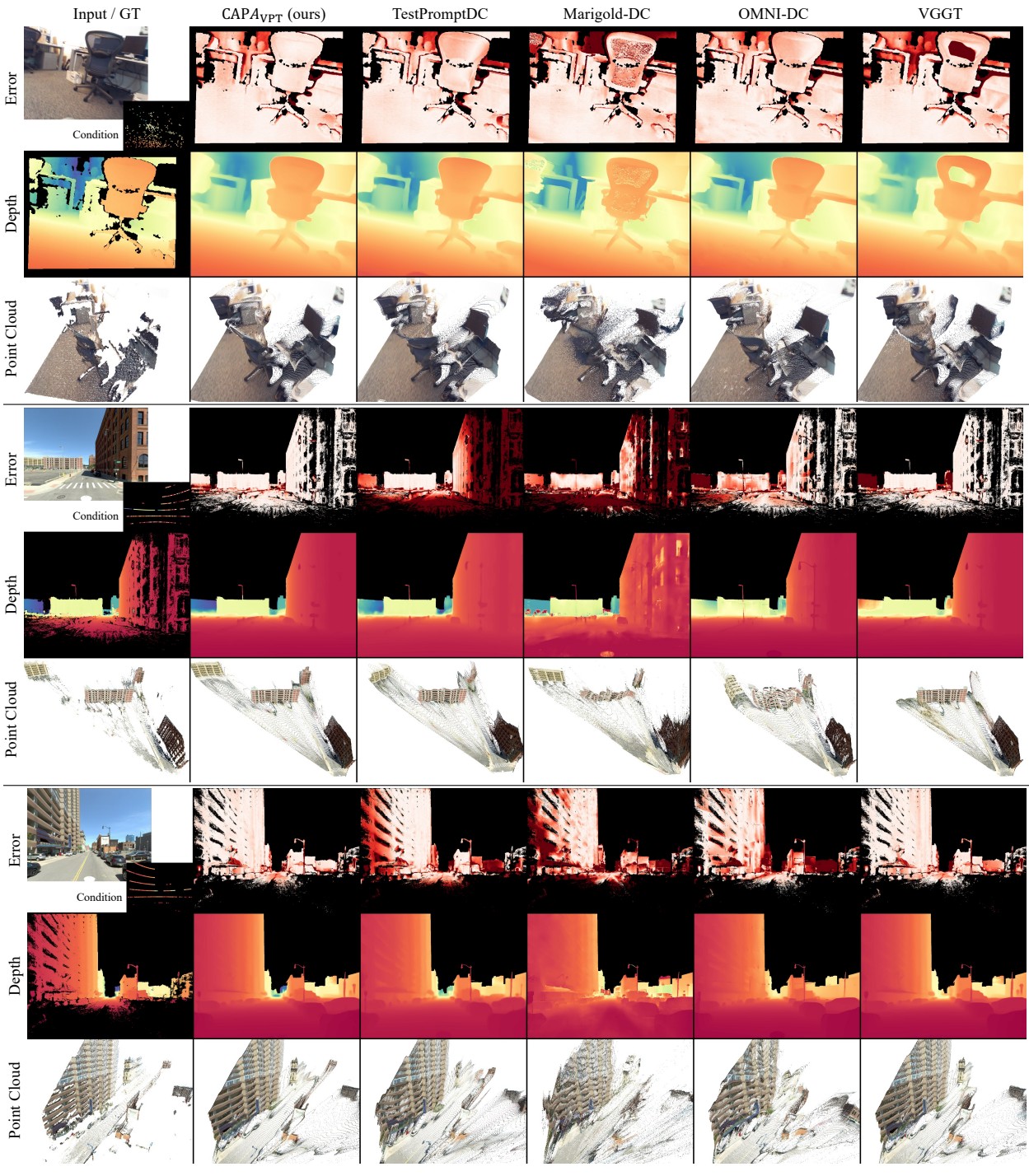

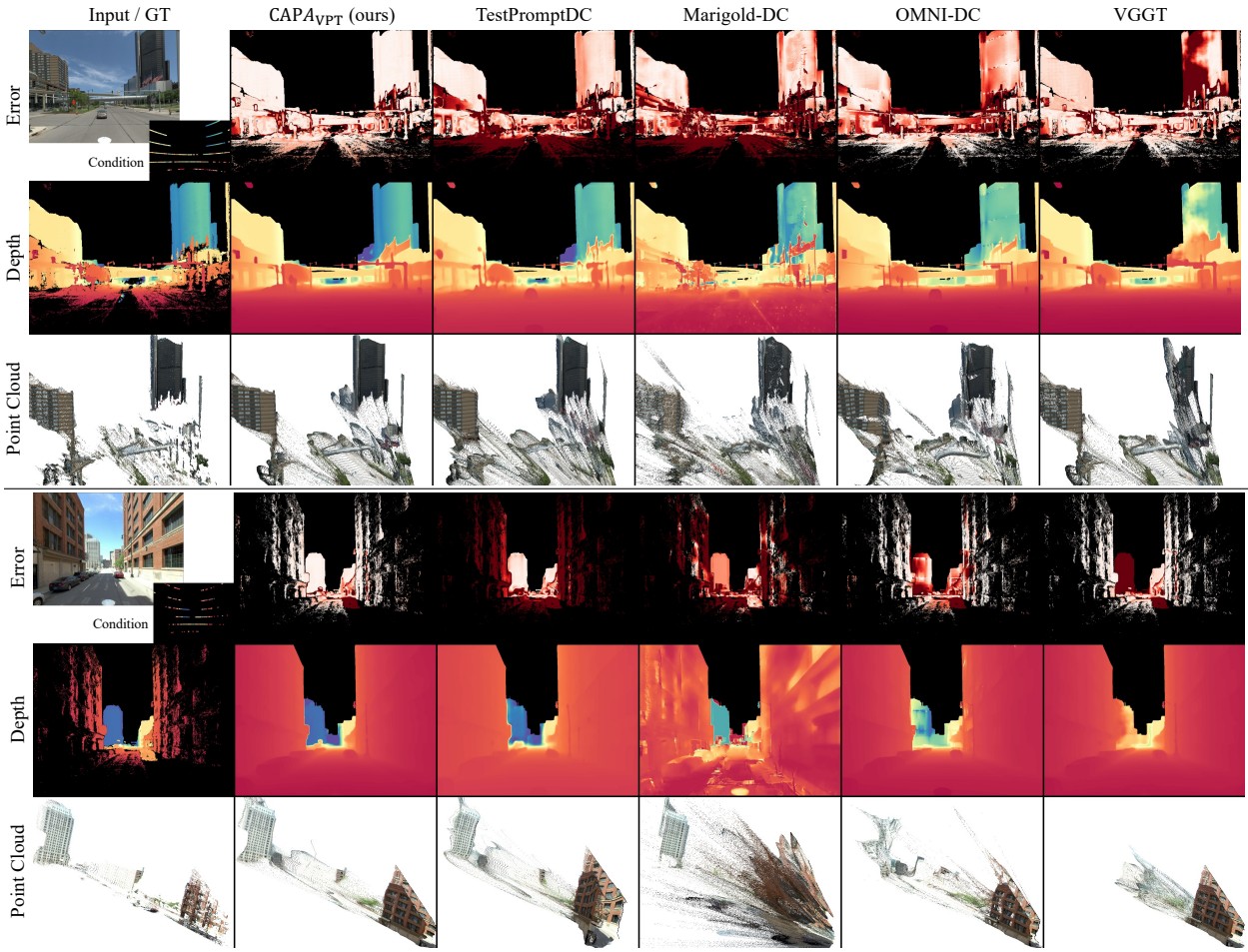

Figure S5: Additional **qualitative comparison on iBims, ScanNet, 7-Scenes, and Metropolis datasets**. Depth is color-coded near ▬ far, errors low ▬ high.

