# OpenReview forum: "Depth Completion as Parameter-Efficient Test-Time Adaptation"
_TMLR — Under review for TMLR_

### Review · Reviewer_HBUk · 2026-04-12

**Summary Of Contributions:**

This paper presents CAPA, a framework that treats Depth Completion as a Test-Time Adaptation (TTA) task. Instead of training task-specific encoders, CAPA leverages the frozen geometric priors of Vision Transformer (ViT)-based Foundation Models (FMs). By optimizing a minimal set of parameters (via LoRA or VPT) using sparse depth observations at inference time, the method aims to ground the foundation model in scene-specific measurements and correct geometric distortions. For video sequences, it introduces a parameter-sharing strategy to ensure temporal consistency.

**Audience:**

Yes

**Audience Explanation:**

The expriments on multiple FMs give audience intersting findings to see how TTA helps FMs on partial obserseved cases.

**Broader Impact Concerns:**

I think it is a engineering adaption for the pratical use, but should focus more on efficiency. While modern foundation models like DA3 are moving towards near real-time performance, CAPA introduces a heavy optimization loop that slows down the process by orders of magnitude. For an engineering adaptation, the authors should focus more on improving efficiency such as combining model distiallation/quantization tricks to make the adpation fast to meet the real world scenarios.

**Claims And Evidence:**

Yes

**Claims Explanation:**

The implmentation and experiments support its claims, although it mainly combines established methods on this depth completion task.

**Requested Changes:**

1.Clarification of Technical Novelty: While using PEFT on Foundation Models is an established technique in many vision tasks, its application as a for depth completion requires more rigorous justification of its novelty.

2.Why is updating the backbone (via LoRA/VPT) necessary compared to simply fine-tuning the output head or a lightweight decoder, or even both? More analysis should be given.

3.When fine-tuning the backbone on a very small set of sparse points (e.g., 100 random points), how does CAPA prevent the model from overfitting to potential noise or outliers in the sparse input? Does updating the backbone's attention maps risk 'breaking' the global structure in favor of satisfying a few noisy local constraints?

4.While CAPA is a pragmatic engineering adaptation for depth completion, the submission lacks a sufficient focus on operational efficiency. Contemporary foundation models, such as DA3, have demonstrated the potential for near real-time, feed-forward performance. In contrast, CAPA introduces a heavy, iterative optimization loop that increases latency by orders of magnitude. For such an adaptation to be truly viable in real-world engineering scenarios (e.g., robotics or AR), the authors should explore efficiency-enhancing techniques such as model distillation, quantization, or accelerated optimization to bridge the gap between high-fidelity adaptation and practical execution speeds.

5.The current evaluation focuses on older or specific base models. To demonstrate that CAPA is truly model-agnostic and state-of-the-art, the authors should provide results or discussions involving more recent Foundation Models such as DA3.

---

> ### Author Response · Authors · 2026-05-09
> **Response from authors (part 1)**
>
> Thank you for the constructive feedback and positive view of our work. We address below the open questions that were pointed out in the review.
> We have revised the manuscript accordingly.
>
> ## Novelty
> - We fully agree that the individual tools used for fitting, i.e. LoRA/VPT, are established techniques, and never claim they were not.
> - Our novelty is to ground 3D foundation models with sparse metric observations, turning depth completion into a scene-specific adaptation task, rather than training a task-specific sparse-depth encoder.
> - CAPA can be applied to different ViT-based depth/3D foundation models with LoRA or VPT, rather than being tied to a particular diffusion pipeline or prompt formulation.
> - For videos, CAPA introduces sequence-level parameter sharing, which aggregates sparse cues across frames and improves robustness to noise and temporal inconsistency without training a new model.
> - Empirically, CAPA achieves marked gains over the frozen base model and over existing TTA depth-completion methods, while updating only a tiny fraction of parameters.
>
> ## Design choice: encoder vs. head
> - We agree that the rationale could be made clearer.
> - Empirically, Tab.6(Tab.7 in revision) compares full-model fine-tuning (FT-All), encoder tuning (FT-Encoder), head tuning (FT-Head), and our encoder-side PEFT variants. Head-only tuning is consistently worse than encoder-side tuning, despite updating substantially more parameters than CAPA. In contrast, LoRA/VPT on the encoder-side attention layers approaches the performance of full fine-tuning while updating only about 0.04% of the model parameters. This also suggests that the improvement is not simply due to adding trainable capacity, otherwise it would not matter so much where the adaptation is applied.
>
> - Our intuition is that biases that need to be corrected are sometimes global rather than local: wrong scale, distorted layout (first example in Fig.3), misplaced structures (first example in Fig.1), or inconsistent geometry across frames (Fig.4). The prediction head mainly decodes the existing representation into dense depth. If the underlying geometric representation is biased, changing only the head has limited ability to reorganize long-range structure. By adapting attention-space parameters, CAPA can modify token interactions and propagate sparse metric cues across the image or sequence, while keeping the original foundation-model weights frozen.
> - In the revision, we have added a short discussion in Sec. 3.1 to make this clearer, and better connected it to Tab.6(Tab.7 in revision).
>
> ## Avoid overfitting to noise in conditioning points
> - We thank the reviewer for raising this important point. CAPA, by its design, mitigates overfitting to sparse or noisy points.
> 1) CAPA does not update the original foundation-model backbone weights. All pre-trained weights remain frozen; only a tiny set of LoRA/VPT parameters is optimized in the attention layers. Thus, the global geometric prior encoded by the foundation model is largely preserved, while the sparse observations provide a lightweight, scene-specific calibration signal.
> 2) The optimization is deliberately constrained: CAPA uses only a tiny part of trainable parameters (e.g., about 0.04% for VGGT), and a small fixed number of adaptation steps. This is much less flexible than full fine-tuning and all but rules out memorization of sparse, noisy points.
> 3) For videos, sequence-level parameter sharing acts as an additional regularizer. The same PEFT parameters must satisfy sparse cues across many frames, so the update is not driven by a few local constraints in one image. This aggregates complementary observations and averages out frame-specific noise/outliers.
> 4) Additionally, we use L1 objectives and gradient clipping during adaptation, which further reduces sensitivity to large outliers.
> - Empirically, our main benchmark already includes noisy sparse observations, following OMNI-DC, where 10% of conditioning points are corrupted. CAPA remains robust in this setting. Table 3 further shows that CAPA has a much smaller error gap between conditioned and unconditioned regions than other depth-completion methods, suggesting it does not simply overfit to the observed points but improves the completed regions as well. Figure S4 also illustrates that CAPA avoids local artifacts around noisy conditioning points better than pixel-space prompt tuning.

---

> ### Author Response · Authors · 2026-05-09
> **Response from authors (part 2)**
>
> ### Efficiency
> - We agree with the reviewer that efficiency is still a limitation, which we also acknowledged in Sec.5 -- while CAPA is significantly faster than existing test-time optimization baselines in our sequence-level runtime comparison, it is still slower than single-pass feed-forward models.
> - We clarify (in the introduction) that CAPA currently targets high-fidelity offline applications such as mapping, reconstruction, and pseudo-groundtruth generation, where strict real-time latency is not critical.
> - Future work will explore methods to accelerate convergence to few steps. The ideas of model distillation or quantization, as proposed by the reviewer, are indeed a promising direction.
>
> ### More recent base model DA3
> - We thank the reviewer for the suggestion. To further test the model-agnostic nature of CAPA, we additionally ran it with the recent DA3 base model under our main protocol.
> - Across 12 settings, CAPA (with both variants) consistently improves DA3. This provides further support to ouor claim that CAPA is not specific to the base models used in the paper.
>
> **Table R5. Quantitative results (AbsRel%) of CAPA based on DepthAnythingV3**
>
> ScanNet
>
> | Method | SIFT | Random 100 | < 3m |
> |---|---:|---:|---:|
> | DAv3 vanilla | 4.0 | 3.4 | 3.5 |
> | DAv3+CAPA_LoRA  | 1.5 | 1.2 | 1.6 |
> | DAv3+CAPA_VPT | 1.5 | 1.1 | 1.5 |
>
> 7-Scenes
>
> | Method | SfM | Random 100 | < 3m |
> |---|---:|---:|---:|
> | DAv3 vanilla | 4.7 | 4.0 | 4.0 |
> | DAv3+CAPA_LoRA  | 3.0 | 1.5 | 1.5 |
> | DAv3+CAPA_VPT | 3.0 | 1.5 | 1.4 |
>
> iBims
>
> | Method | SIFT | Random 100 | < 5m |
> |---|---:|---:|---:|
> | DAv3 vanilla | 3.9 | 3.2 | 3.7 |
> | DAv3+CAPA_LoRA  | 2.8 | 2.3 | 2.9 |
> | DAv3+CAPA_VPT | 3.0 | 3.1 | 2.9 |
>
> Metropolis
>
> | Method | 8-line | 16-line | 32-line |
> |---|---:|---:|---:|
> | DAv3 vanilla | 26.4 | 27.1 | 26.0 |
> | DAv3+CAPA_LoRA  | 18.6 | 16.9 | 16.4 |
> | DAv3+CAPA_VPT | 19.5 | 17.1 | 16.5 |

---

### Review · Reviewer_4m7c · 2026-04-27

**Summary Of Contributions:**

+ The paper devices a method to estimate depth with sparse observations during test time, unlike other (expensive) train time methods.
+ It aligns scale with appropriate formulation
+ It uses PEFT and prompts to come up with adaptation of parameters to learn depth
+  It simply uses high fidelity foundation models and adapts them
+ At the level of the sequence, parameters are shared leveraging learning across 'similar' scenes.

I view this as an interesting bridge between few shot learning and test time adaptation, and would work well in many settings (e.g. indoor settings). I am curious about autonomous driving settings where depth estimation in this manner would be very interesting.

Metrics:
 I found the OPW metric interesting - warp to next frame with flow and measure error. The claim is that the other methods 'overfit' to sparse observations and show increased error elsewhere.

The benefit of using sequence level (versus frame level) adaptation is shown with an ablation showing reduced error and improved temporal consistency.

Weaknesses
---------------
+ I think none of the fittings used are 'novel' as such, so this is an incremental work (but quite interesting and relevant)
+ Heavy reliance on foundation model
+ Sequence level overfitting. It seems to me that the method will not generalize to new sequences. We have to present it examples from a sequence to calibrate it.
+ It is not sample efficient - from my reading of the plots it takes 100s of samples for the error to go down.

**Additional Comments:**

None

**Audience:**

Yes

**Audience Explanation:**

Depth estimation is an important problem in computer vision. Practitioners will be interested in developments in this space.

**Claims And Evidence:**

Yes

**Claims Explanation:**

The paper is founded on sound scientific reasoning and the method is demonstrated with solid metrics.

**Requested Changes:**

Runtime and compute - the 100s of samples figure jumped out when I saw it. I think it should be a main figure rather than appendix.
Test protocol - please clarify if for adaptation one needed to see the entire sequence or just samples for the experiments.
Failure modes - please clarify what the failure modes are. I am interested in the sparse/dense transition. Are there regions where the adaptation fails? I think this should be systematized and ablated.

---

> ### Author Response · Authors · 2026-05-09
> **Response from authors**
>
> We thank the reviewer for the thoughtful comments and positive appraisal of our work. We address below the limitations and open questions mentioned in the review.
> We have revised the manuscript accordingly.
>
> ## Test protocol for sequences
> - In our sequence evaluation, the full test sequence is available and each sequence is evaluated independently -- no information is shared across different test sequences.
> - Baselines that do not support sequence-level adaptation are run frame-by-frame on the same evaluated frames, using the authors' official protocols.
> - CAPA uses this sequence for test-time adaptation by sampling mini-batches of frames at each optimization step. As shown in Fig.6, we empirically found that a 10% frame mini-batch in each optimization step is sufficient while improving efficiency.
>
> ## Autonomous driving settings
> - We have outdoor/autonomous-driving-style evaluations on Metropolis with LiDAR-like scan-line patterns (Tab.1, last qualitative example in Fig.3, and Fig.4), as well as KITTI and DDAD in Tab.S6.
>
>
> ## Reliance on foundation model
> - This reliance is intentional: CAPA is designed to make strong foundation models conditionable by sparse metric observations, rather than replacing them. As foundation models improve, CAPA directly benefits from that progress.
> - We also show consistent gains on VGGT, MoGe-2, and UniDepthV2, suggesting the method is not tied to a specific backbone.
>
> ## Novelty
> - We of course agree that the individual fitting tools, e.g., LoRA/VPT, are not new.
> - The novelty lies in formulating depth completion as test-time PEFT grounding of 3D foundation models with the help of sparse metric cues.
> - CAPA can be applied to different ViT-based depth/3D foundation models with LoRA or VPT, rather than being tied to a particular diffusion pipeline or prompt formulation.
> - The sequence-level extension is a key contribution: one shared set of adapted parameters is optimized across frames, exploiting inter-frame correlation to improve robustness and temporal consistency.
> - Empirically, CAPA achieves marked gains over the frozen base model and over existing TTA depth-completion methods, while updating only a tiny fraction of parameters.
>
> ## Sequence-level overfitting
> - CAPA’s sequence-specific adaptation is intentional, it is a feature not a bug.
> - In practical depth completion, each sequence can have different scale, sensor sparsity, camera motion, and scene geometry. Rather than hoping for a fixed model to cover all such cases, CAPA uses the sparse measurements available at test time to calibrate the frozen foundation model to the specific sequence.
> - Sharing PEFT parameters across frames reduces single-frame overfitting by aggregating cues from multiple views, improving robustness and temporal consistency.
>
> ## Efficiency and runtime table
> - Thanks for the suggestion, we have moved the runtime table (Tab.S11) to the main paper (Tab.5 in revision).
> - We have also made the accuracy-cost trade-off more visible. Fig.7 shows that even a small number of optimization steps improves over the aligned base model, with performance saturating around 100 steps. Hence, the user has an intuitive tuning nob to trade accuracy for runtime.
>
> ## Failure mode
> - The typical failure mode is discussed in Sec.C10. We have extended Conclusion for the connection to Sec.C10.
>
> - Failures mainly occur when the base foundation model fails, e.g., on mirror/glass/non-Lambertian surfaces, or when a wrongly reconstructed reason is not cov ered by sparse observations. In such cases, CAPA does not receive the necessary corrective signal to recover the missing geometry.
>
> ## Sparse-dense transition
> - We revised and made Sec.C4 easier to find from the main text. Fig.S1 and Tab.S7 show that sequence-level adaptation is strongest in very sparse regimes, while per-image adaptation can benefit more once sufficiently dense conditioning points are available.

---

### Review · Reviewer_zcPJ · 2026-04-27

**Summary Of Contributions:**

* **Strenghs**

- **Competitive adaptation performance on the depth completion benchmarks.**

- **Competitive multi-view optical-flow based warping error.**

- **Improved adaptation speed compared to the previous baseline.**

* **Weaknesses**

- **Limited novelty**.

- **Lack of information regarding the implementation**.

- **Lack of justification and analysis to update the encoder**.

- **Unfair evaluation metric on temporal consistency**.

- **Use of the relative evaluation metrics as the main result**.

- **Unfair temporal consistency metric (OPW)**.

- **Lack of investigation of the affine transformation.**

**Additional Comments:**

.

**Audience:**

No

**Audience Explanation:**

The problem of depth completion is a well-established problem, which predicts the metric depth from a single image and its synchronized depth, enoughly appealing to the audience.

A problem of multi-view geometry is also on the classic geometry learning, by estimating the camera pose and the correspondence between the points. However, the way the authors implemented the adaptation of the foundation depth estimation model with the Parameter-Efficient Fine-Tuning and multi-view sounds incremental and redundant to the reviewer.

Mainly the multi-view aspect of the proposed method comes from the adaptation with the multi-view inputs, where the most of the baselines except the VideoDA and VGGT. Especially, none of the depth completion baseline is trained on the multi-view.

Also, the combination of the two for the adaptation is contradictory and the realization of the idea is somewhat redundant. If VGGT already outputs the metric depth, why does the proposed method first linearly shift and scale the prediction and update the model on the sparse depth? Could it be called depth completion or just an adaptation of the multi-view geometry model to the sparse observation? Is the affine transformation computed on a single frame or on the entire sequence? If low-rank tuning or visual prompt-tuning is not sufficient to cover the scale ambiguity of the depth estimation models, it may not be the effective way to adapt the model.

**Claims And Evidence:**

No

**Claims Explanation:**

**Limited novelty**: The reviewer is not fully convinced by the novelty of the proposed contribution. The core idea appears to be a replacement of the pixel-wise visual prompt tuning with well-known parameter-efficient finetuning/adaptation methods, i.e., Low-Rank Adaptation and Visual Prompt Tuning, for depth completion by finetuning the model on sparse depth observations available at test time. The idea of linear scale-shift is analogous to the previous TestPromtDC paper. While the sequence-level adaptation is useful, the reviewer finds that the paper should more clearly distinguish the methodological novelty from a straightforward application of existing PEFT techniques to depth estimation/completion.

**Limited clarification on evaluation**: While a prior method, TestPromptDC, also adapts a foundation monocular depth estimation model at test time, the authors evaluate the proposed method on a substantially different set of datasets and settings. Moreover, the quantitative results reported for TestPromptDC do not appear to clearly align with the numbers in the original TestPromptDC paper. Since TestPromptDC provides an official implementation, the reviewer suggests that the authors reproduce the method using the official code and report the official/reproduced numbers under a clearly matched evaluation protocol.

**Insufficient evidence to support model-agnostic performance**: The authors claim that the proposed Parameter-Efficient Fine-Tuning framework is model-agnostic. However, the main quantitative evaluation is primarily centered on VGGT, while the additional results on other base models are relatively limited. Since the proposed method is claimed to be broadly applicable to different ViT-based depth/3D foundation models, the reviewer believes that more comprehensive demonstrations across multiple depth estimation backbones are needed to support the model-agnostic claim.

**Metric-depth evaluation metrics should be shown in the main paper**: The problem of depth completion is fundamentally to estimate metric depth. Therefore, reporting only the relative error metric, i.e., Absolute Relative Error, in the main paper is less convincing from the reviewer’s perspective. The reviewer suggests moving metric-scale depth estimation metrics, such as Mean Absolute Error and Root Mean-Squared Error, from the appendix to the main paper.

**Unfair comparison of the OPW metric**: The reviewer is concerned about the fairness of the OPW-based temporal consistency comparison. Depth completion is originally a single-view problem, while the proposed sequence-level adaptation uses multi-view/video frames during the adaptation stage. In contrast, several baselines are not given the same multi-frame adaptation access. Therefore, the comparison may not be fully justified. The reviewer suggests separating single-frame and sequence-level evaluations and comparing each method under equivalent temporal input and adaptation assumptions.

**Contradictory optimization procedure of the method**: The proposed method first aligns the prediction using scale and shift estimated by L1 minimization between the prediction and valid sparse depth pixels. This suggests that the estimated scale and shift can change depending on the sampling pattern and spatial distribution of the sparse depth observations, which is less convincing to the reviewer. If the assumption of linear alignment is sufficient, then it is unclear why low-rank optimization is necessary. Conversely, if linear alignment is insufficient, then the method should analyze the sensitivity of the proposed optimization to the sparse-point distribution. The reviewer suggests including ablations such as affine-only alignment, no-alignment adaptation, and robustness to different sparse-depth sampling patterns. There is also a recent study showing that linear alignment may not hold reliably for metric depth estimation [a].

**Lack of analysis on updating only the encoder**: The proposed method only updates the encoder. The authors state that “by updating only the attention layers within the ViT, we can guide the model’s global geometry to align with sparse observations, avoiding the need to fine-tune the dense decoder or the full backbone.” However, the reviewer finds this explanation mostly heuristic. How do the authors analyze that global geometry emerges specifically in the encoder and can be aligned by updating only encoder-side attention layers? Is there any empirical or prior evidence supporting this design choice? In addition, why do the authors not include PEFT variants applied to the decoder or to both encoder and decoder? The current paper does not provide sufficient justification for selecting encoder-only parameter-efficient finetuning.

**Requested Changes:**

* The reviewer suggests adding an ablation without scale-and-shift alignment. Since the proposed method estimates scale and shift from sparse valid pixels before computing the adaptation loss, it is unclear how much of the improvement comes from affine calibration versus PEFT-based adaptation. The authors should report comparisons among affine-only alignment, PEFT without scale-and-shift, and the full method. In addition, it would be useful to analyze whether PEFT parameters adapted on one scene transfer to other scenes with similar geometric or scale statistics in target dataset (may include similar scale and shift), or whether the learned parameters are strictly scene-specific.

* The reviewer suggests the experiment needs to clarify the separated inference time to compute scale-shift and updating the model. What is the main bottleneck of adaptation? Computing the scale-and-shift parameter computation or PEFT?

* The review suggests the authors to present more analysis on the rationale behind updating only the encoder with PEFT. Is there any correlation between the estimated depth and encoder feature or decoder feature? How is the result if the model decoder is trained with PEFT?

* The reviewer suggests to move the metric depth error metrics (MAE and RMSE) to the main table than the relative error, or putting them together if the authors would like to.

* Elaborate the main difference between the TestPromptDC results in the main paper and the original paper.

---

> ### Author Response · Authors · 2026-05-09
> **Response from authors (part 1)**
>
> We thank the reviewer for the detailed comments and for carefully examining the evaluation and design choices of our method. We appreciate the opportunity to clarify and address the concerns.
> We have revised the manuscript accordingly to incorporate these points.
>
> ## Novelty and relation to existing TTA methods
> - We agree that LoRA/VPT themselves are established PEFT techniques, and we obviously do not claim them as novelties, we simply use them as best-practice tools. Our novelties are:
> 	-  CAPA adapts internal attention-space parameters of the foundation model, while TestPromptDC injects pixel-space prompts. This difference is important because sparse depth cues often require correcting global geometric structure, such as layout, scale-related distortions, and misplaced objects, rather than only local input perturbations. We also empirically found that this design is more robust to noise in the conditioning.
> 	-  CAPA extends TTA depth completion **from frame-wise adaptation to sequence-level** adaptation via shared PEFT parameters, enabling multi-frame cue aggregation and improved temporal consistency. TestPromptDC and other TTA methods like Marigold-DC are essentially single-frame and do not provide a sequence-level adaptation.
> 	-  CAPA is broadly applicable to ViT-based 3D/depth foundation models. Note that CAPA applies to **both multi-view and single-frame** models and can also improve the consistency of single-frame models. We demonstrate it on VGGT, MoGe-2, and UniDepthV2.
> 	- These design differences lead to empirical advantages: CAPA is more robust to very sparse or noisy conditions, it achieves higher accuracy and better temporal consistency in our experiments, and it is also faster than TestPromptDC or Marigold-DC when applied to sequences.
> - The affine scale-shift alignment is a standard process to align depth maps, widely used in the literature (see discussions below). We definitely do not claim it as our contribution, we simply follow the standard protocol for consistency. We will make this clearer in the revision.
>
> ## Evaluation protocol and datasets
> - We thank the reviewer for pointing out that this part could be made clearer.
> - Our goal is to evaluate depth completion under both single-image and sequence settings. Therefore, the benchmark includes single-image datasets such as iBims, video/RGB-D datasets such as ScanNet and 7-Scenes, and outdoor sequence data such as Metropolis.
> - For fairness, all methods are evaluated with **the same RGB inputs and the same sparse depth observations** under each dataset/condition setting. No information is shared across different test sequences. Methods that do not support sequence-level inference or adaptation are run frame-by-frame, using the authors' recommended protocol. CAPA’s sequence-level variant is evidently evaluated as a per-sequence method.
> - This distinction is clarified in Sec. 4.1 "Baselines". We have revised and made the presentation clearer by separating frame-level and sequence-level settings more explicitly..

---

> ### Author Response · Authors · 2026-05-09
> **Response from authors (part 2)**
>
> ## Comparison with TestPromptDC
>
> - We have reproduced TestPromptDC using the authors' original, official inference code under our unified evaluation protocol, where methods are evaluated with the same RGB inputs, sparse-depth inputs, noise settings, and metrics within each dataset/setting.
> - The difference from the original TestPromptDC numbers are due to the fact that they were not computed with the exact same protocol, e.g., there can be differences in terms of the sparse-depth patterns, the noise added for noisy settings,, the resolution (for NYU), and preprocessing of the dataset. We summarize these differences in Table R1.
> - To isolate the effect of the evaluation protocol, we additionally report MAE (%) under an aligned setting, without injecting noise, for the overlapping datasets in Table R2. For iBims and NYU, we match TestPromptDC's random GT-point densities, while keeping the original resolution of NYU. Since the original sparse masks are unavailable, minor differences may remain due to resampling randomness, NYU resolution, and implementation details. For VOID and DDAD, we use the available sparse-depth settings provided with the dataset: 1500-point sparse depth for VOID and LiDAR sparse depth for DDAD.
> - Under this aligned TestPromptDC-style setting, CAPA remains comparable to or better than TestPromptDC across the overlapping datasets, consistent with the trend in our unified noise-free benchmark (Tab.S4).
>
> **Table R1. Protocol differences and TestPromptDC results (MAE%, lower is better) on overlapping datasets.** Each TestPromptDC result is reported under the protocol shown in the same column; these rows are intended to quantify differences in evaluation protocol, they are not apple-to-apple comparisons.
>
> | Dataset | TestPromptDC paper | CAPA protocol | Aligned setting here |
> |---|---|---|---|
> | iBims | 1000 random GT points; TestPromptDC: 4.3 | 100 random GT points, with/without 10% noise; reproduced: 16.1 (Tab.1) / 5.7 (Tab.S4)| 1000 random GT points, noise-free; reproduced: 4.3 |
> | NYU | 500 random GT points, lower resolution; TestPromptDC: 4.1 | 100 random points, with 10% noise; reproduced: 13.0 (Tab.S6) | 500 random GT points, noise-free; reproduced: 2.8 |
> | VOID | 1500 sparse points; TestPromptDC: 13.2 | dataset-provided 150 / 500 sparse points; reproduced: 4.2 / 3.4 (Tab.S6) | dataset-provided 1500 sparse points; reproduced: 3.4 |
> | DDAD | random 20% sparse depth; TestPromptDC: 148.2 | dataset-provided LiDAR sparse depth; reproduced: 156.2 (Tab.S6) | dataset-provided LiDAR sparse depth; reproduced: 156.2|
>
> **Table R2. Comparison under and aligned setting without injected noise (MAE%, lower is better)**.
>
> | Method | iBims | NYU | VOID | DDAD |
> |---|---:|---:|---:|---:|
> | (Condition pattern) | random 1000 GT points | random 500 GT points | dataset-provided 1500 points | dataset-provided LiDAR |
> | TestPromptDC (reproduced) | 4.3 | 2.8 | 3.4 | 156.2 |
> | CAPA_LoRA | 3.3 | 2.7 | 3.3 | 129.8 |
> | CAPA_VPT | 3.4 | 2.9 | 3.4 | 141.0 |
>
>
> ## Model-agnostic evidence
> - We use **3 different base models** (VGGT, MoGe-2, UniDepthV2), independently developed in distinct research groups. All three are popular recent models and representative of the state of the art. Specifically, VGGT is the prototypical **multi-view** geometry model, while MoGe-2 and UniDepthV2 are both **single-frame** models. The results are presented in Fig.5, Tab.S2, Tab.S3, Tab.S4, and Tab.S5, showing **consistently improved** accuracy and temporal consistency.
> - We revised and made these results more visible in the main text and clarify that our method is model-agnostic **within the family of ViT-based depth/multi-view foundation models.**
> - Additionally, we have also included the results based on DA3 in the reply to reviewer `HBUK`
>
> ## Metric depth evaluation metrics (AbsRel clarification)
> - There seems to be a misunderstanding: AbsRel $= \frac{|\hat{d} - d|}{d}$ is computed between a **metric prediction** $\hat{d}$ and a **metric ground truth** $d$; it is therefore a metric-scale evaluation metric, not a scale-invariant relative-depth metric.
> - AbsRel is a popular evaluation metric used in much of the recent literature on depth estimation (e.g., Marigold, DepthAnything, UniDepth) and depth completion (e.g. OMNI-DC, PriorDepthAnything).
> - We also reported MAE and RMSE in the appendix (Tab.S2-S4).
> - The **conclusions drawn from these metrics are consistent** and suggest that, with different base models, CAPA consistently outperforms both the respective base model and other SOTA depth-completion methods.
> - Given the comparatively large number of datasets, conditioning patterns, and methods, including all three metrics would in our view overload the main table and hurt readability. We therefore kept AbsRel in the main table and report MAE/RMSE comprehensively in the appendix.
> - We have added the detailed definitions of the evaluation metrics as Sec.B4.

---

> ### Author Response · Authors · 2026-05-09
> **Response from authors (part 3)**
>
> ## Temporal consistency and OPW fairness
> - We emphasize that OPW is used only as a post-hoc evaluation metric: all methods are evaluated on the same frames using the same optical-flow warping protocol. Thus, the  computation of the metric is uniform. It is only applied to sequence datasets.
> - OPW is a standard temporal-consistency metric (or loss) in video depth estimation, e.g., in NVDS (ICCV'23), RollingDepth (CVPR'25), Buffer Anytime (CVPR'25), CH3Depth (CVPR'25), VideoDA (CVPR'25)
> - The temporal-access assumption is separate from the metric itself. The difference lies in each method’s supported inference mode. CAPA is explicitly designed to support sequence-level adaptation, while several depth completion baselines are single-frame methods, so the best we can do is run them frame-by-frame, using the authors' recommended, official protocols. Sequence-level parameter sharing is an integral part of CAPA’s contribution, so we do not see why the evaluation should be unfair.
> - We never claim that all methods would use the same temporal information internally. Rather, the OPW table evaluates the temporal consistency of the depth maps produced by each method, whatever its supported inference mode. This is important because one message of CAPA is that sequence-level parameter sharing is beneficial in terms of both accuracy and consistency.
> - To make this clearer, we have revised the manuscript to explicitly indicate which methods are single-frame, video/multi-view.
>
> ## Affine alignment and model adaptation
> - We point out that VGGT does **not** provide guaranteed metric-scale geometry: In Sec. 3.4 *“Ground Truth Coordinate Normalization,”* Wang et al. note that *scaling a scene does not affect the input images, and they resolve this ambiguity by normalizing camera translations, point maps, and depth maps using the average Euclidean distance of the 3D points*. Thus, VGGT is trained to predict a canonical normalized scale rather than an externally calibrated metric scale.
> - In our method, the affine alignment and optimization serve two different but complementary purposes:
> 	1. Affine alignment brings the prediction to metric scale with a **coarse global alignment**, serving as a fast calibration baseline that quickly removes global scale/shift errors. and the starting point of optimization.
> 	2. The optimization, which is the main focus of our method, repairs **scene-level distortion and local errors** beyond linear alignment, using the geometric priors from the base models.
> - Given non-metric prediction and metric reference (GT points or, in our case, conditioning points), using an affine transformation is a standard practice to align the depth maps (c.f. MiDaS, Marigold, DepthAnything).
> - Affine alignment alone is **not** sufficient and the optimization step is essential, as shown in our results:
> 	- E.g., for the quantitative comparisons (Tab.1-2, Tab.S2-S5), the baseline methods are aligned to conditional points, without optimization. With CAPA plugged in, the accuracy and temporal consistency improve significantly.
> 	- In the supplementary website, videos under the "Optimization Process" section visualize the optimization process, where "step 0" (i.e., the starting point of the error curves) corresponds to the results after affine alignment, before optimization. The curves directly show the improvement due to optimization
> 	- alignment uses the L1 objective (see Eq.4), it is therefore not overly sensitive to large noise. Our empirical results with noisy condition points empircally confirm  the robustness.
> - In our implementation, affine scale/shift is estimated per frame from that frame’s sparse observations -- a natural way to use the available input; only the PEFT parameters are shared across frames.
> - We provide additional ablations where we remove global alignment and use scale-only alignment. These results show three consistent trends, across different base models.
>     1) removing global alignment significantly degrades performance, even for metric models, both with and without CAPA, confirming that metric calibration from sparse observations is important for non-metric or imperfectly scaled base predictions.
>     2) scale-only and affine alignment perform similarly overall, although affine is usually slightly better, indicating that CAPA is not too sensitive to the exact parameterization used for global alignment.
>     3) CAPA improves over the aligned base model under the corresponding scale-only or affine setting, showing that the gains come from PEFT-based adaptation, not merely from global calibration.
>     - We report representative MAE (%) results on the abaltion subset below for selected conditioning patterns.

---

> ### Author Response · Authors · 2026-05-09
> **Response from authors (part 4)**
>
> ## (continue) Affine alignment and model adaptation
> **Table R3. Ablation of global alignment and CAPA adaptation.** MAE(%) under selected conditioning patterns; lower is better. `none`, `scale`, and `affine` denote no alignment, scale-only alignment, and affine scale-shift alignment, respectively.
>
> | Method | ScanNet SIFT | 7-Scenes SfM | iBims SIFT | Metropolis 8-line |
> |---|---:|---:|---:|---:|
> | VGGT (none)| 118.5 | 98.8| 233.6| 5284.9 |
> | VGGT (scale) | 5.8 | 10.5| 16.3 | 914.9|
> | VGGT (affine)| 5.8 | 10.6| 14.1 | 916.4|
> | VGGT+LoRA (none) | 30.1| 28.8| 50.0 | 1502.6 |
> | VGGT+LoRA (scale)| 2.3 | 6.1 | 4.8| 466.6|
> | VGGT+LoRA (affine) | 2.3 | 5.7 | 5.4| 478.0|
> | VGGT+VPT (none)| 39.8| 58.5| 57.9 | 1954.6 |
> | VGGT+VPT (scale) | 2.5 | 5.5 | 4.5| 498.7|
> | VGGT+VPT (affine)| 2.5 | 4.9 | 4.6| 508.3|
> | MoGe-2 (none)| 28.4| 13.6| 37.5 | 1537.5 |
> | MoGe-2 (scale) | 9.3 | 7.2 | 11.8 | 1420.4 |
> | MoGe-2 (affine)| 8.4 | 7.3 | 10.0 | 1364.4 |
> | MoGe-2+LoRA (none) | 4.3 | 6.3 | 5.1| 684.7|
> | MoGe-2+LoRA (scale)| 3.0 | 5.3 | 6.2| 626.8|
> | MoGe-2+LoRA (affine) | 2.8 | 5.4 | 6.3| 624.1|
> | MoGe-2+VPT (none)| 5.4 | 7.1 | 6.6| 734.2|
> | MoGe-2+VPT (scale) | 3.3 | 5.7 | 5.2| 674.2|
> | MoGe-2+VPT (affine)| 3.2 | 5.6 | 5.0| 668.2|
> | UniDepthV2 (none)| 14.9| 19.5| 28.0 | 2189.2 |
> | UniDepthV2 (scale) | 7.6 | 8.2 | 11.9 | 1448.2 |
> | UniDepthV2 (affine)| 6.9 | 8.0 | 10.6 | 1450.0 |
> | UniDepthV2+LoRA (none) | 4.3 | 6.6 | 5.2| 664.0|
> | UniDepthV2+LoRA (scale)| 3.0 | 5.8 | 6.2| 623.7|
> | UniDepthV2+LoRA (affine) | 2.9 | 5.6 | 7.0| 617.8|
> | UniDepthV2+VPT (none)| 5.0 | 7.1 | 6.2| 707.5|
> | UniDepthV2+VPT (scale) | 3.1 | 5.9 | 5.3| 640.5|
> | UniDepthV2+VPT (affine)| 2.9 | 5.9 | 6.8| 632.6|
>
> - The citation marked as "[a]" was not included in the review text, so we are unable to identify that specific study. We would be happy to consider it if the reviewer can provide the reference.
> - Regarding **runtime** of these two parts: the affine alignment is much cheaper than test-time optimization, since it only solves for two numbers based on the sparse depth correspondence; whereas the LoRA/VPT optimization requires forward and backward passes. We provide the runtime breakdown of the two parts below.
>
> **Table R4. Runtime (second) breakdown of alignment, optimization, and final inference.** Note: the total runtime has ~3% difference compared to Tab.S11(Tab.5 in revision), due to normal system factors such as CUDA scheduling, memory allocator/cache state, GPU contention, thermal or power variability.
>
> | Method    |   Affine Alignment Time |   PEFT Optimization Time |   Final Inference Time |   Total Time |
> |-----------|-------------------------|--------------------------|------------------------|--------------|
> | CAPA-LoRA |                    1.5 |                   144.4 |                   9.1 |       155.0 |
> | CAPA-VPT  |                    1.7 |                   151.2 |                   8.9 |       161.8 |

---

> ### Author Response · Authors · 2026-05-09
> **Response from authors (part 5)**
>
> ## Encoder-side PEFT vs. head tuning
> - We thank the reviewer for raising this point. We agree that it makes sense to explain the rationale for encoder-side adaptation in more detail.
> - Our current ablation in Tab.6(Tab.7 in revision) already compares different adaptation locations through full fine-tuning: FT-Encoder, FT-Head, and FT-All. The results show that head-only tuning is consistently **worse** than encoder-side tuning. FT-All gives only marginal gains over FT-Encoder/CAPA, despite updating far more parameters. Based on these findings, we decided to focus on encoder-side PEFT, which approaches the performance of full fine-tuning while updating only about 0.04% of the model parameters. This also suggests that the improvement is not simply due to adding trainable capacity, otherwise it would not matter so much where the adaptation is applied.
> - The intuition is that one must often correct the geometry globally, for instance in case of wrong scale, distorted layout (first example in Fig.3), misplaced structures (first example in Fig.1), and temporal consistency. The decoder/head mainly maps the existing representation to dense depth. If the underlying geometric representation is biased, adapting only the head has limited ability to reorganize long-range structure. Encoder-side attention PEFT can instead modulate token interactions and propagate sparse metric cues across the image/sequence, while keeping the original foundation-model weights frozen.
> - In the revision, we have added a short discussion in Sec. 3.1 to make this clearer, and better connected it to Tab.6(Tab.7 in revision).
>
> ## Task formulation: depth completion vs. model adaptation
> - CAPA’s input/output is still depth completion: RGB + sparse metric depth → dense metric depth.
> - The sequence version is video depth completion, not a different task.
> - CAPA also works on single-frame models, so it is not merely VGGT/multi-view adaptation.
>
> ## Scene-specific adaptation
> - This scene-specificity is intentional, a main strength of CAPA is to adapt the generic prior to the specific setting: different sequences can have different metric scales, sensor sparsities, camera motions, and scene layouts. Re-optimizing lightweight PEFT parameters at test time allows CAPA to calibrate to these differences without retraining the foundation model.
> - Cross-scene generalization ability comes from the frozen foundation model; PEFT variables are optimized and discarded/reused only for the same sequence.
>
> ## Implementation details
> - The reviewer notes that some implementation details are missing, but the comment does not specify which component is unclear.
> - Our manuscript describes the main adaptation and evaluation settings in Sec. 4.1 and Appendices A/B, including LoRA/VPT settings, optimizer, learning rates, adaptation steps, frame mini-batching, sparse-point sampling, and noise injection.
> - If the reviewer could point out any specific implementation detail that remains unclear, we will be happy to address it directly in the revision.
> - Additionally, the code will be released open-source to guarantee reproducibility of every implementation detail.

---

### Comment · Action_Editor_CkdT · 2026-06-29

Dear Authors,

There seems to be outstanding concerns. Some of which seem addressable within a short time. Can you comment on the points below?

Thanks,

AE

====

1. Rationale for Encoder-Only PEFT

The request for further analysis on why only the encoder is updated via PEFT was not directly addressed. To clarify, the reviewer requested an experiment where the decoder is fine-tuned with PEFT—not full-parameter fine-tuning. Is there a specific correlation between the estimated depth and the encoder versus decoder features that justifies this design choice? The reviewer would still like to see the results of a PEFT-trained decoder.

2. Depth Error Metrics

The reviewer reiterates the request to move the metric depth error metrics (MAE and RMSE) to the main table, rather than just relative error (or putting them together, if the authors prefer). Regarding the authors' clarification on AbsRel: Absolute Relative Error (AbsRel) is inherently a relative metric, as the physical units in both and cancel out after division. Its name explicitly confirms that it measures relative, not metric, error.

3. Fairness in Temporal Consistency Comparisons

While the reviewer is well-acquainted with the OPW and temporal consistency metrics, the primary concern lies in the fairness of the baseline comparisons. The problem of depth completion is predicting a metric depth map from a single-view image and its synchronized sparse measurement, not optimizing multi-view jointly.

Comparing CAPA (which is evaluated on multi-view images and temporal prediction consistency) against other models optimized purely for single-view inputs is unfair. In this context, the only strictly valid model to compare against in the manuscript is VideoDA. Given that using the sparse observation condition is applicable to other methods (e.g., VideoDA), the authors could further improve the comparisons by adding these condition-based optimizations (e.g., ProxyTTA) alongside scale-and-shift alignment (e.g., TestPromptDA) to the baseline monocular/multi-view depth estimation models as well.

Acknowledged Rebuttals & Additional Notes

The reviewer acknowledges that the following comments from the previous review stage were adequately addressed in the rebuttal:

Affine Alignment and Model Adaptation: "The reviewer suggests adding an ablation without scale-and-shift alignment. Since the proposed method estimates scale and shift from sparse valid pixels before computing the adaptation loss, it is unclear how much of the improvement comes from affine calibration versus PEFT-based adaptation. The authors should report comparisons among affine-only alignment, PEFT without scale-and-shift, and the full method. In addition, it would be useful to analyze whether PEFT parameters adapted on one scene transfer to other scenes with similar geometric or scale statistics in target dataset (may include similar scale and shift), or whether the learned parameters are strictly scene-specific." (Addressed)

Inference Time: "The reviewer suggests the experiment needs to clarify the separated inference time to compute scale-shift and updating the model. What is the main bottleneck of adaptation? Computing the scale-and-shift parameter computation or PEFT?" (Addressed)

Comparison with TestPromptDC: "Elaborate the main difference between the TestPromptDC results in the main paper and the original paper." (Addressed)

Additional Comment on Affine Alignment: The reviewer is knowledgeable that scenes predicted by foundation depth completion models can suffer from distortion, mainly from Out-Of-Distribution, and acknowledges that there could be confusion on the point previously raised. Still, the affine alignment does not make much sense to the reviewer, as the scale and shift parameters predicted by a given condition
 could be different under different conditions, whereas the underlying image inputs remain the same. This concept exists in ProxyTTA (CVPR 2024), but the fundamental difference here is whether the network is given the sparse depth input to observe.

---

> ### Author Response · Authors · 2026-06-29
> **Authors' reply to Action Editor CkdT**
>
> Dear Action Editor,
>
> Thank you for your message and for summarizing the remaining concerns.
> We appreciate the opportunity to clarify these points, and respond below with the revisions we propose to make.
>
> ### 1. Rationale for Encoder-Only PEFT
>   - Design choice justification:
>
>     a. The full-parameter fine-tuning ablation (Tab.7) shows that tuning the **decoder is already worse** than tuning the encoder.
>
>     b. Since decoder-side PEFT has even **less trainable capacity**, we do not expect it to even reach the full-parameter tuning results or change the conclusion.
>
>     c. The effect of sparse conditioning points is not purely local:
>       -  The decoder, e.g. the DPT head, mainly relies on convolutional local/multi-scale processing --> local impact only
>       -  while the ViT encoder attention blocks are better suited for long-range token interaction and global geometric correction.
>       - For example, in the first row of Fig. 1, the building is corrected as one coherent large structure from only three rows of sparse conditioning pixels.
>
>   - If the AE still considers decoder-side PEFT essential after this clarification, we will run this experiment. A fair decoder-side PEFT comparison would require a few days and additional GPU resources for thorough hyper-parameter search, e.g. adapter placement, rank, learning rate, and optimization steps.
>
> ### 2. Depth Error Metrics
>   - **We can add MAE in addition to AbsRel to the main table (Tab.1)**.
>   - However, we would like to note that
>     - the conclusions drawn from AbsRel / MAE / RMSE in this paper are consistent;
>     - having all three metrics in one table will result in a **very small font size, making the table unreadable**;
>     - AbsRel is also an important metric, widely used for depth estimation and completion tasks.
>
> ### 3. Fairness in Temporal Consistency Comparisons
>   - Clarification:
>
>     a. Regarding task formulation:
>       - For **single-image** evaluation, we have included **iBims**, where **CAPA achieves better results** (Tab.1, S2-4).
>       - We further **extended the evaluation to video sequences** with more datasets.
>
>     b. For methods that do not take conditioning points as input (e.g. VideoDA), we **applied scale-shift alignment** using the same conditioning points before evaluation.
>
>     c. On sequence datasets, for methods that are only designed for single images, we did not modify their input interface, but ran them frame-by-frame, and ensured that they receive the same conditioning points where applicable.
>   - We have classified baselines in Sec. 4.1 "Baselines" by input types. We **will indicate this more clearly** also in the main table.
>   - We would appreciate **clarification on what additional comparison** would best address the remaining concern. In particular, do you suggest implementing a sequence-level variant of TestPromptDC as additional baseline?
>
> Looking forward to your reply.
>
> Best regards,
>
> Authors

---

> > ### Comment · Action_Editor_CkdT · 2026-06-30
> >
> > Dear Authors and zcPJ,
> >
> > For clarification, the summary was provided by a reviewer and it seems that they believe the points to be important to address.
> >
> > Regarding the first point, the reviewer's request seems to be pointed towards analysis based on the question "Is there a specific correlation between the estimated depth and the encoder versus decoder features that justifies this design choice?". Additionally, PEFT may avoid parameter drift and can exhibit a different behavior than full finetuning. This may be a fair point to address and can strengthen the manuscript.
> >
> > Regarding the second point, the AE agrees with the author that adding both may overcrowd the table and one, either MAE or RMSE, is sufficient. Amongst the two, the AE would recommend RMSE as MAE would seem somewhat redundant given AbsRel.
> >
> > Regarding the third point, "In this context, the only strictly valid model to compare against in the manuscript is VideoDA. Given that using the sparse observation condition is applicable to other methods (e.g., VideoDA), the authors could further improve the comparisons by adding these condition-based optimizations (e.g., ProxyTTA) alongside scale-and-shift alignment (e.g., TestPromptDA) to the baseline monocular/multi-view depth estimation models as well."
> >
> > Can Reviewer zcPJ clarify the exact baseline they are looking for? Please list the steps that you expect the authors to take to address this point.
> >
> > Thanks,
> >
> > AE